# The Development of the Optimal Harvesting Model of an Offshore Fishery Supply Chain Based on a Single Vessel

Ming-Feng Yang [1,2], Sheng-Long Kao [1,2,3] and Raditia Yudistira Sujanto [2,*]

1 Department of Transportation Science, National Taiwan Ocean University, Keelung City 202, Taiwan; yang60429@mail.ntou.edu.tw (M.-F.Y.); slkao@mail.ntou.edu.tw (S.-L.K.)
2 Intelligent Maritime Research Center, National Taiwan Ocean University, Keelung City 202, Taiwan
3 Center of Excellence for Ocean Engineering, National Taiwan Ocean University, Keelung City 202, Taiwan
* Correspondence: sujanto.raditia@gmail.com

**Abstract:** This study delves into the offshore fishing industry in Taiwan, emphasizing the significance of the aquatic product market, supply chain development, and the maturity of cold chain technology. Taiwan's geographical advantage of being surrounded by the sea provides a thriving environment for marine resources and migratory fish. This study is motivated by the increasing demand for diverse fish products, driven by the growing need for high-quality protein. Recognizing the importance of meeting this demand, this study aims to investigate the capacity of logistics systems and cold storage in the offshore fishery industry, particularly under conditions of uncertainty. To tackle the optimization challenges prevalent in the offshore fishery supply chain, this study employs the bat algorithm (BA), a metaheuristic algorithm inspired by the remarkable echolocation behavior of bats. Additionally, a systematic literature review methodology is utilized to gather relevant articles and establish a comprehensive understanding of the study domain. This study culminates in proposing an optimized fishing model for the offshore fishery supply chain, highlighting the significance of evaluating supply chain value from a management perspective and identifying existing deficiencies and bottlenecks in current research. By focusing on optimizing the offshore fishery supply chain, this study aims to enhance the industry's efficiency and effectiveness, providing valuable insights and recommendations to improve the capacity of logistics systems and cold storage. Furthermore, this study presents the results of the BA, showcasing its effectiveness in approaching optimization challenges, thereby validating its utility for the offshore fishery industry. Sensitivity analysis reveals the potential for higher profits by raising the inventory limit of the manufacturer, enabling the supplier to provide materials to more profitable trading partners. While this study is based on a revenue and cost model, it acknowledges that the objectives and constraints would become more complex in varying logistic system circumstances. The future study aims to expand the scale of the model and incorporate practical cases to further enhance its applicability.

**Keywords:** supply chain; offshore fishery; nonlinear programming; bat algorithm; inventory limit; profit maximization

## 1. Introduction

The Taiwanese offshore fishing industry has experienced significant growth in recent years, with the aquatic product market becoming export oriented. This development has led to the maturation of the product supply chain, supported by advancements in cold chain technology. The supply chain encompasses crucial processes such as harvesting, product processing, and the transportation of raw materials and finished products. These processes are influenced by the dynamic environment, particularly the demands arising from both local and global markets [1]. The increasing demand for various fish products is driven by the growing need for high-quality protein. Moreover, wild fishing activities are dependent on factors such as fish species, weather conditions, and government regulations [2]. Despite

the abundance of marine resources, several challenges persist, impeding the efficiency of the offshore fishery supply chain. These challenges can be categorized into two main areas: (a) the dynamic nature of the environment, including fluctuating demands in local and global markets, which directly impacts fishing, production, and distribution activities within the offshore fishery industry; and (b) supply and demand issues, encompassing the delicate balance between fish species availability, weather conditions, and compliance with government regulations. These factors collectively affect the stability and efficiency of the supply chain. Nonetheless, the offshore fishing industry plays a vital role in generating economic and social impacts, while also providing inherent scientific benefits. Ma et al. [1] argue that studying the capacity of logistic systems and cold storage under conditions of uncertainty is crucial. However, existing studies have primarily focused on the flow and supply chain of pelagic fishery, leaving the offshore fishery supply chain relatively unexplored. This gap highlights the need for further investigation in order to improve the understanding and management of the supply chain processes specifically related to the offshore fishing industry.

Constructing a profit-maximizing model for the offshore fishery industry is necessary for optimizing the supply chain, meeting market demand, managing resources, and addressing complex supply chain problems. Furthermore, it assesses the impacts of industry development on cost problems and fishery management, improving coordination and efficiency. In this study, it is assumed that refrigerated storage is exclusively applicable to offshore fishing within the model. This study considers the diverse temperature requirements for different fish species and environmental conditions, and thus utilizes a fixed temperature setting based on the suitable temperature for the majority of species. Additionally, this study assumes that only vessels from local offshore fishing companies are accepted, and variables in the model include the cost of racks, construction expenses, and other influential costs. Prior studies have attempted to identify the various influential factors in the supply chain where purchased quantity, fish weight, sale price, inventory, and transportation are identified as the most influential factors in the whole supply chain model [3–5]. However, these studies focused on different assumptions such as how three types of supply chain are able to coexist and interact with each other, how information disclosure might infer the demand received by the downstream players, and how carbon footprint is considered and estimated in the capture seafood industry management. This study, nevertheless, focuses on proposing a profit-maximizing model to reach an efficient supply chain management in the offshore fishery industry.

A supply chain includes issues of management and coordination, as well as the relationship between the various participants in the supply chain. In addition, the supply chain operation, involving the entire activities and decisions from suppliers to consumers, is essential to a set of fishery supply chain models because the model is critically dependent on the environment and the use of natural resources [2,6]. Moreover, conducting a supply chain analysis involves inspecting the operation of the various stakeholders, any deal between stakeholders in the supply chain, the influence among the stakeholders, and the stakeholders' relationship evolution. From the perspective of management, Rosales et al. [7] have emphasized the importance of connections among the stakeholders, which potentially augmented the supply chain value. Nevertheless, it is necessary to evaluate the potential impacts of the industrial chain development, from upstream to downstream, on cost problems and fishery management.

Thus, this study focuses on optimizing a fishing model for the offshore fishery supply chain. On the one hand, the supply chain offers problems that cannot be solved by using linear programming. There is a wide range of approximate methods to solve such problems using various metaheuristic algorithms. For instance, the BA is inspired by the echolocation behavior of bats when searching for prey in nature. The BA is widely used due to its simplicity, ease of handling, and applicability to various problems [8]. The method is effective for solving continuous optimization challenges [9]. In addition, nonlinear programming is applied using Python. The study's objectives are as follows:

- To develop a model of supply chain based on profit maximization;
- To identify the crucial player in the offshore fishery supply chain;
- To provide decision-makers with a solution based on the determining parameters in the offshore fishery supply chain.

This study contributes to the literature by developing an optimal model that enhances the offshore fishery supply chain through an identification of the important player in the supply chain. The result of this study demonstrates the effectiveness of the BA in tackling the optimization challenges of the offshore fishery supply chain, providing valuable insights for enhancing the industry's efficiency and effectiveness. The managerial implications include providing a solution for the decision-makers in Taiwan's offshore fishery supply chain.

The rest of the study is organized in five sections of which Section 2 presents the literature review in relation to supply chain and the fishery supply chain model. Section 3 explains the materials and methods applied in the study. The results are discussed in Section 4 which is divided into theoretical and managerial implications. Lastly, Section 5 concludes the study and presents the study's limitations and a direction for future study.

### 1.1. Literature Review of Supply Chain

Supply chain has an extendable definition. Prior studies have associated the term with a supplier providing products to a single buyer or multiple buyers [10,11]. A supply chain is a chain of processes involving manufacturers, suppliers, retailers, and consumers, which have an inventory system with multi-retailer with a continuous review function, that maximizes profits to optimize costs and inventory management [12–15]. Desiderio et al. [16] provided a narrow perspective in describing the supply chain by limiting the range of activities performed within a firm to produce a certain output that ends at the consumer. From a broad perspective, supply chain is defined as a process that starts from the production system of raw materials and moves along the linkages with other sectors and firms engaged in trading, harvesting, processing, manufacturing, pricing, transporting, advertising, and others; the performance is enhanced by implementing logistical strategies [2,17]. Tliche et al. [4] suggested that these strategies include capacity utilization, reduced quantity of inventory, increased flexibility, improved responsibility to customer needs, reduced delivery times, and increased system and process transparency.

Further, Aragão et al. [5] claimed that a supply chain is the management of the entire production process, starting from processing, distributing, and ultimately purchasing activities by consumers. Supply chain includes a wide range of aspects such as risks, costs, quantity, capability of transportation, and inventory [3,6,18]. Thus, there is consensus among studies that the supply chain entails the processes that involve a firm's operational activities that are affecting and affected by the other stakeholders along the process.

### 1.2. Offshore Fishery Supply Chain Model

In the context of the offshore fishery industry, the supply chain begins with producers or fishermen and terminates with end-buyers who sell to consumers. End-buyers include retail outlets, that is locally owned fish markets to supermarket chains and restaurants. The objective functions and indicators of the design of the fishery supply chain could include a wide range of aspects, like risks, costs, quantity, capability of transportation, and inventory, which might involve multiple levels with diverse products during different periods [4,18,19]. Offshore fishery products are under the categorization of perishable products that are subject to spoilage and are short-lived, causing a challenge in maintaining the quality, and require a specific harvesting and distribution model [20,21]. For example, the supply chain needs to consider a process of temperature control that is a vital need of cold chain transportation and storage affecting an increase in energy costs. Prior studies have attempted to develop different models with the integration of inventory model, coordination between a single buyer and a single seller, and economic order quantity [10,22]. Yang et al. [23] comprehensively studied the cost and energy consumption of

perishable products in the production, transportation, and sales process and formulated integrated marketing strategies on the basis of pricing, temperature, and quality control. Bányai et al. [24] described a heuristic optimization mathematical model for sequential supply based on a flower pollination algorithm, focusing on aspects of sustainability, including fuel consumption and emissions, including allocation and scheduling issues.

Studies have identified the potential incomes and costs in this supply chain model [25,26]. For instance, manufacturing costs such as packaging, freezing, labor costs, and transportation costs as well as both deterioration and maintenance costs would be considered in this model. Moreover, Song et al. [26] contributed a network flow model based on integrating problems of cold storage location as well as yield uncertainty and demand dynamics for distant-water fishery supply chains. The complexity of the supply chain processes is depicted in Figure 1 that provides a visual overview to illustrate the complex relationships in the network, containing the process of fishing in the fishing grounds, harvesting by fishery vessels, and the process of transportation to manufacturing plants, through storage in manufacturer/wholesaler cold storage and delivery to the final consumers/retailers through these distribution channels.

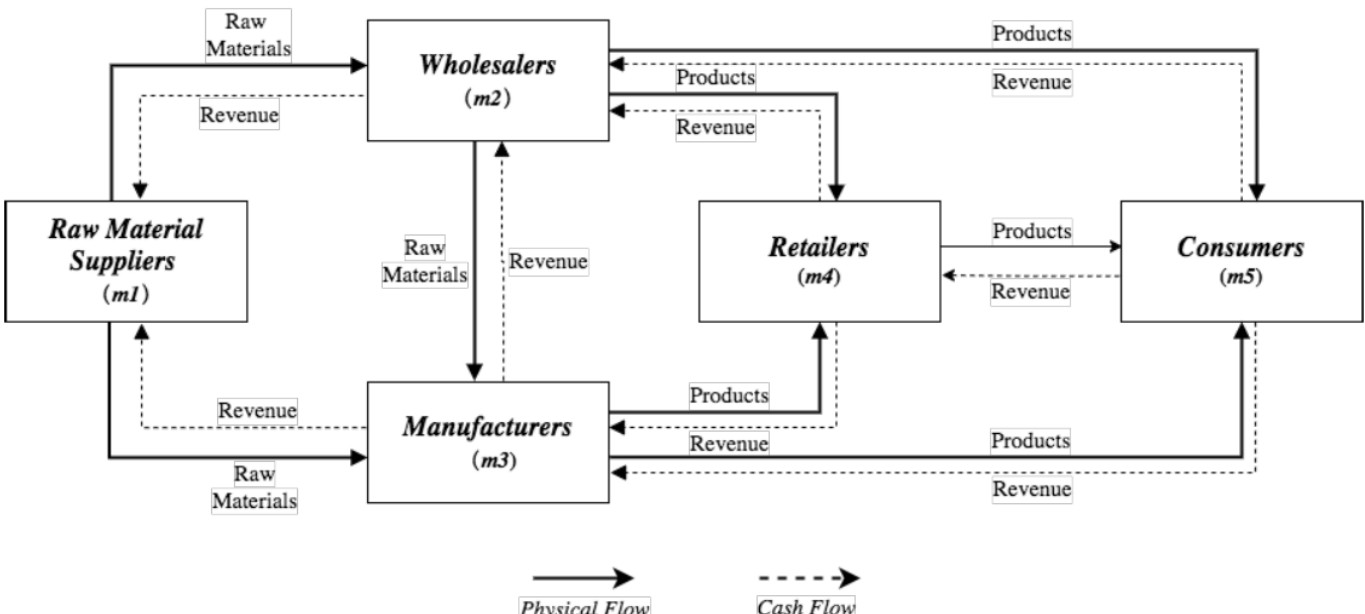

**Figure 1.** Conceptual framework for integrated logistics.

Compared with the traditional supply chain, the factors considered in the fishery supply chain are different in terms of side of production (fishery production) and supply (seasonal fish species). For this kind of perishable product industry, operation costs in the supply chain of perishable aquatic products are rather high due to a dilemma in cold-chain transport and storage. When allocating the capacity of cold storage and inventory storage of manufacturers and fish-processing plants, decisions in offshore fishery industry chains should be deliberated. Yang et al. [23] proposed a model by considering energy consumption and the cost involved in the process of freezing, transportation, and trading. The impacts of perishable products for energy consumption on strategies of replenishment and management, deduction of costs, and improvement of resource utilization were analyzed. Ma et al. [1] inferred that decision variables, such as depth, number of rows, level, and aisle, of the storage capacity were affected by the harvesting effort, storage capacity of plants, batch size, and productivity. Structure variables such as height and length were used to allocate the cold storage capacity of the port. Moreover, Liu et al. [27] considered the optimal freezing storage capacity of offshore fisheries under uncertain fishing (or production)

conditions and proposed an optimization model for both the optimal refrigeration capacity and maximal expected results.

This study assumes that refrigerated storage is only applicable to offshore fishing in this model. Since different species of fish require different temperatures for cold storage and condition of environments, according to the suitable temperature setting for most species, cold storage with a fixed temperature setting has been used. Moreover, this study assumes the accepted vessels are only those of local offshore fishing companies and considers the set cost of racks, construction cost, and other influential costs as variables in the model. Overall, the assumptions are compiled in Table 1.

### 1.3. Previous Studies

The BA has been known as a formidable optimization tool and is notable for its productivity in the solution of law-dimensional functions, and a different range of applications [8]. For instance, the method was used to optimize the operating costs of thermal power plants; however, it has been indicated that as the scale of the problem increases, the capability of the BA might be diminished [8]. Thus, maintaining good performance with the increasing complexity of the problem is a difficult task using the method. Nevertheless, an ensemble BA was proposed to solve large-scale optimization problems and introduced integration ideas.

Compared with other excellent swarm intelligence optimization algorithms, the superiority of the BA is confirmed for solving large-scale optimization problems. There is an improved version of the original BA, known as the fractional Lévy flying bat algorithm in which the velocity is updated as the score calculation and the local search process with random walks based on the Lévy distribution progress. Effectively, this change improves the ability of the algorithm to get rid of local solutions. Furthermore, a novel hybrid model was introduced, incorporating the Lagrangian Relaxation (LR) technique, a metaheuristic method, the BA, and a practical Swarm Optimization method. This model aims to address the complex long-term production scheduling problem arising from deterministic assumptions and various levels of uncertainties. Through its integration of diverse optimization strategies, this hybrid model offers a near-optimal solution to the aforementioned problem.

**Table 1.** Selection of literature review based on the assumptions and methods.

| Reference | Assumptions | Methods |
|---|---|---|
| Macusi et al. [2] | The study assumes that shrimp aquaculture is facing the need to increase production to meet growing food demand, particularly in the Philippines where there has been a shift from milkfish to prawn due to market demand. It acknowledges the positive and negative impacts of intensification on the environment and socioeconomic aspects. | The study utilizes the PRISMA method to review and assess the challenges faced by the shrimp aquaculture industry, including environmental issues like farm management, marine pollution, disease outbreaks, and social, economic, and climate change impacts. |
| Flynn et al. [3] | The study assumes the existence of three types of supply chain uncertainty, namely, micro-level, meso-level, and macro-level, and hypothesizes that these uncertainties coexist and interact with each other, while also positing that supply chain integration, centralization, formalization, and flatness influence the dimensions of uncertainty, which are tested using hierarchical regression analysis. | A hierarchical regression analysis based on data collected from 339 manufacturing plants. |

**Table 1.** *Cont.*

| Reference | Assumptions | Methods |
|---|---|---|
| Tliche et al. [4] | The study assumes that utilizing a downstream demand inference (DDI) strategy with various forecasting methods, including the proposed weighted moving average, can effectively address information disclosure issues in decentralized supply chains and improve supply performance metrics, while formalizing the manufacturer's forecast optimization problem. | A comprehensive approach encompassing downstream demand inference (DDI) strategy, various forecasting methods, analysis of demand processes, evaluation of supply performance metrics, formalization of the manufacturer's forecast optimization problem, and application of Newton's method for solution. |
| Aragão et al. [5] | The study assumes that greenhouse gas emissions are crucial for climate change mitigation, and that emission assessments are typically excluded from the management of capture fisheries. It also assumes that hake is an important food in Spain and aims to estimate the carbon footprint of the hake seafood chain by analyzing its extraction, transport, and distribution. | The analysis of the hake seafood chain in Spain, which includes the extraction (fishing), international transport, and domestic distribution within Spanish territory. The data is obtained from various sources, such as vessels' logbook records for hake landings, the official database of seafood trade flows, and the Food Consumption Panel data. |
| Desiderio et al. [16] | The study assumes that achieving sustainable production and consumption in food systems requires considering the entire supply chain and the social dimensions that are often overlooked compared to environmental and economic aspects. It acknowledges the lack of consensus and measurement methods regarding social sustainability in food supply chains. | The study reviews 101 papers and gray literature documents to identify the current state of measuring social sustainability using various tools and indicators. The analysis focuses on five stages of the food supply chain (production, processing, wholesale, retail, and consumer) and four stakeholders (farmers, workers, consumers, and society). |
| Karray and Martín-Herrán [17] | This research assumes that the introduction of a store brand in a supply chain with competing national brands can have significant impacts on the profitability of different supply chain members, particularly the national brand manufacturers. The study focuses on analyzing the effects of manufacturers' decision timing choices regarding pricing and advertising on the competition between national and store brands. | A game-theoretic model is developed, considering different decision timing scenarios. |

Despite the dimensionality and constraint issues, this study addresses some remedies. Prior studies have used the BA and proved that the method was effective in achieving the well-known constraint benchmark solution [28]. Yılmaz and Küçüksille [29] emphasized the use of an enhanced BA and adopted the capability of the standard test functions and constrained problems. Nevertheless, the different contexts where the BA was applied included a control of energy consumption and tracking accuracy in robots [30], an exploration of buried ore and mineral targets parameters [31], and an assessment of multi-classification problem of Cardiotocography [32]. Meanwhile, the application of the algorithm in the context of offshore fishery supply chain is scarce in the literature. Moreover, while it is true that the models used in this study are nonlinear, the claim regarding the feasibility space being a convex set and the existence of a global optimal solution is found in prior studies that applied nonlinear models to find an optimal solution [1,28,29]. Although nonlinearity typically complicates the analysis, certain techniques and mathematical frameworks can be employed to verify convexity properties and establish the existence of global optima.

In sum, these studies resulted in proving that the enhanced BA is a superior method, as illustrated in Figure 2.

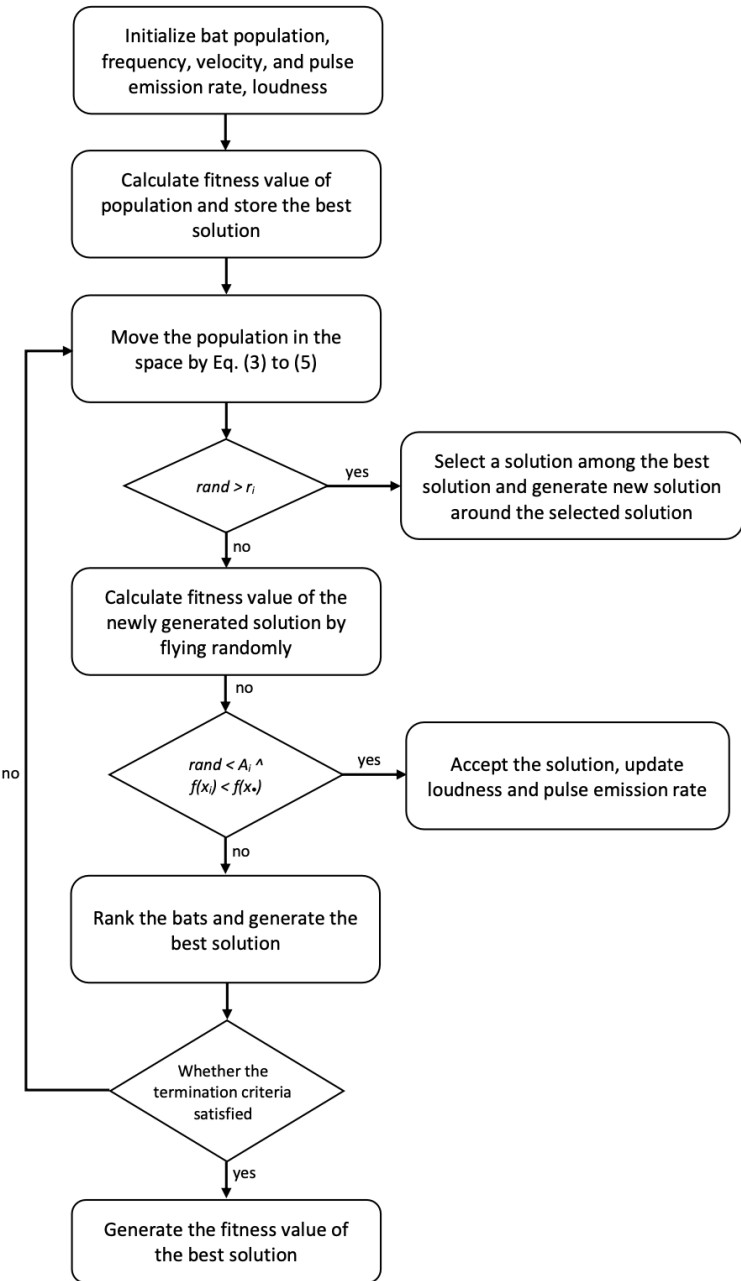

**Figure 2.** Flowchart of enhanced bat algorithm.

## 2. Materials and Methods

This section reviews the methodology used in prior studies, explains the BA, its structure, formulation of the optimization problem, and disaggregates inventory constraints.

### 2.1. Review Methodology

The kind of coordinated relationship involving strategies, relationships, and activities in the supply chain integrates diverse participants from the chain, which include suppliers, manufacturers, distributors, and retailers, to achieve efficiency and deliver value to customers. The methodology of systematic literature review includes (a) finding the definition of the study's topic; (b) searching for the origins of literature such as Scopus and Web or Google Scholar, then narrowing the number of articles while identifying the article's main concept; (c) defining the methodology of the selected articles; (d) describing the results of the optimization problems in the articles; and (e) indicating the deficiencies and bottlenecks

of the current study. For defining the relevant keywords, it is crucial to review articles in diverse fields of supply chain management and aim for the articles with the most similar concept. The keywords that are used to search for the articles based on the titles, abstracts, and keyword lists in the internet database include "supply chain model" "optimizing model", and "optimal bat algorithm". The literature introduces a wide range of methods used to solve optimizing problems under a single vessel considering the offshore fishery supply chain.

## 2.2. Bat Algorithm

Metaheuristic algorithms such as Genetic Algorithm, Particle Swarm Optimization, and Ant Colony Optimization have become powerful methods for solving many tough optimization problems, whose characteristics are described in Table 2.

**Table 2.** Characteristics of different metaheuristic algorithms.

| Metaheuristic Algorithms | Characteristics |
|---|---|
| Bat Algorithm [9,33] | • The BA replicates how bats would act if they were randomly seeking prey, in a manner inspired by the way that they use echolocation.<br>• Bats utilize echolocation to find their prey and travel. Similar to this, each bat in the BA stands for a potential solution, and the frequency and loudness of the bats indicate how good the solution is.<br>• Bats use ultrasonic pulses to exchange information with one another and to improve their communication by varying their frequency and loudness.<br>• In order to explore the search space, the BA uses local and global search tactics. It also adjusts the frequency and loudness of bats to go towards the best solution. |
| Genetic Algorithm [34] | • Natural selection and evolution are the sources of the Genetic Algorithm's inspiration. It reflects how evolution works in nature, where the strongest survive.<br>• Each member of the population used by the Genetic Algorithm represents a potential remedy for the issue.<br>• It employs genetic operators including selection, crossover, and mutation to produce novel progeny and raise population standards across many generations.<br>• Positive qualities are more likely to spread through time as a result of selection for reproduction favoring individuals with greater fitness (i.e., those that are closer to the ideal solution). |
| Particle Swarm Optimization [35] | • Swarm optimization is based on how a collection of people (particles) behave collectively in a search space.<br>• This algorithm mimics the actions of a swarm in which each particle represents a potential answer.<br>• Particles move around the search area and modify their placements depending on their own experience and the experience of nearby particles in Swarm Optimization methods like Particle Swarm Optimization.<br>• Particle Swarm Optimization seeks to converge towards the optimal solution by balancing exploration (global search) and exploitation (local search) of the search space.<br>• The movement of particles is driven by their current position, velocity, and the best solution discovered by any particle in the swarm. |
| Ant Colony Optimization [36] | • The foraging habits of actual ants looking for food served as the basis for Ant Colony Optimization.<br>• Ant Colony Optimization makes use of an artificial ant colony, where each ant represents a potential route to a solution.<br>• The pheromone concentrations have an impact on the likelihood that other ants will choose the pathways that ants leave behind as pheromone trails on the terrain they investigate.<br>• The algorithm employs positive feedback, where ants reinforce the path with higher pheromone concentrations when they find better solutions.<br>• Ant Colony Optimization aims to find the shortest path or optimal solution by exploiting the accumulated pheromone information and balancing exploration and exploitation of the search space. |

According to Vu-Huu et al. [37], one of the key advantages of the BA is its robust capability for auto-zooming. As the search process approaches global optimality, the pulse emission rates and variations in loudness intensify, allowing the BA to focus on promising areas with potential solutions. This auto-zooming ability gives the BA a competitive edge over other metaheuristic approaches. Additionally, the BA differs from many other metaheuristics in terms of parameter control. Instead of using fixed parameters determined by the algorithm, the BA adjusts its parameters based on the values of loudness and pulse emission rates. This adaptive parameter control enables the BA to automatically switch from exploration to exploitation when optimal results are achieved.

In comparison to Genetic Algorithm, Swarm Optimization, and Ant Colony Optimization, the BA is especially helpful in discovering new solutions because of its ability to balance exploration and exploitation of the search space, adaptability, diversity of solutions, and capability to handle constraints. Utilizing local and global search tactics for exploitation, the BA strikes a balance by adjusting frequency and loudness to explore uncharted territory. Due to its dynamic alterations made possible by its adaptive processes, it is able to adapt to the shifting nature of the problems it faces. The algorithm's adaptability and unpredictability encourage a variety of solutions, delaying convergence and allowing for more thorough investigation. Furthermore, because it can manage constraints well, the BA is a good choice for restricted optimization issues. However, the selection of an algorithm should be based on the features of the issue, the resources at hand, and empirical analyses utilizing a variety of performance measures to evaluate performance in various circumstances.

Among the methods, the BA was introduced based on the echolocation ability guiding the foraging behavior of miniature bats [38]. By observing the behavior and characteristics of microbats, the used idealized rules in the BA as a method include:

- All bats use echolocation to sense distance and $x_i$, the location of a bat, and encode the position as a solution to the optimization problem under consideration.
- Bats fly with velocity $v_i$ at position $x_i$ with a frequency $f_{min}$, varying wavelength $\lambda$ and loudness $A_0$ to search for prey randomly. They automatically adjust the wavelength (or frequency) of their emitted pulses and regularly the rate of pulse emission r $\in [0, 1]$, based on the proximity of the prey source.
- The loudness ranges from a large (positive) $A_0$ to a minimum value (constant) $A_{min}$ as assumed in this study.

*2.3. Bat Algorithm Structure*

a. Initialization of bat population: The search space is assumed as a region that contains many preys in it. The algorithm attempts to find the high-quality or best position in the search space. Since the locations of the prey source are unknown, the initial population is randomly generated from real-valued vectors with dimension $d$ and number $N$, by taking into account lower and upper boundaries. Then, the prey source located within the population are evaluated.

$$x_{ij} = x_{min} + \varphi(x_{max} - x_{min}) \tag{1}$$

where $i = 1, 2, \ldots, N, j = 1, 2, \ldots, d$, and $x_{max}$ and $x_{min}$ are upper and lower boundaries for dimension $j$, respectively. $\varphi$ is a randomly generated value ranging from 0 to 1.

b. Generation of frequency, velocity, and new solution: The movements could be influenced by the evaluated fitness values of all bats. For each bat $(i)$, its position $(x_i)$ and velocity $(v_i)$ in a d-dimensional search space should be defined. $x_i$ and $v_i$ should be updated subsequently in the iterative process. The rules for updating the position and velocities of a bat $(i)$ are given below:

$$f_i = f_{min} + (f_{max} - f_{min})\beta, \tag{2}$$

$$v_i^t = v_i^{t-1} + (x_i^t - x_*)f_i, \tag{3}$$

$$v_i^t x_i^t = x_i^{t-1} + v_i^t \tag{4}$$

where $\beta \in [0,1]$ is a random vector derived from a uniform distribution. Here, $x_*$ is the current global best location (solution), which is determined after comparing all results among n bats. As the product $\lambda_i f_i$ is the velocity increment, this study uses $f_i$ (or $\lambda_i$) to adjust the velocity while fixing the other factor $\lambda_i$.

c.  Local search capability of the algorithm: In order to improve the local search capability of the algorithm, once a solution is selected among the current best solution, a new solution will be generated locally using a random fly for each bat.

$$x_{new} = x_{old} + \varepsilon A^t, \tag{5}$$

where $x_{old}$ is an effective solution chosen by some mechanism (e.g., roulette wheel), $A^t$ is average loudness value of all bats at $t_{th}$ time step, and $\varepsilon$ is generated randomly ranging from $-1$ to 1.

d.  Loudness and pulse emission rate: The loudness $A_i$ and the rate $r_i$ of pulse emission have to be updated according to the iteration processing. When a bat approaches its prey, the loudness $A$ and the pulse emission rate r will be updated. Pulse emission rate $r$ is increased as loudness $A$ is decreased with respect to Equations (6) and (7) respectively.

$$A_i^{t+1} = \alpha A_i^t, \tag{6}$$

$$r_i^{t+1} = r_i^0 [1 - \exp(-\gamma t)], \tag{7}$$

where $\alpha$ and $\gamma$ are constants. For any value, $0 < \alpha < 1$ and $\gamma > 0$, where $\alpha$ and $\gamma$ are constants. $r_i^0$ is the initial pulse emission rate value of the $t$th bat.

The first step of the BA is the initialization numbers of bats, and each bat is determined by initial position (solution), random pulse rate, loudness, and frequency. During the iteration, all bats will shift from the initial position to the best position, that is, the global best solution. After movements, if any bat finds a better solution, the bat will update the pulse emission rate and loudness. During the flight iteration process, the best solution will be updated.

These processes will stop until the termination criteria are reached. A pseudo-code developed to solve the model optimization problem with the BA, is shown in Figure 3.

---

**Standard Bat Algorithm**

1 Initialize bat population $x_i$ and velocity $v_i$
2 Define Frequency and updating velocity
3 Initialize pulse emission rate $r_i$ and loudness $A_i$
4 **repeat**
5     Generate new solutions by adjusting frequency and updating velocity and location by Eqs. 2 to 4
6     **if** *rand* $> r_i$ **then**
7         Select a solution among best solutions
8         Generate new local solution around selected best solution
9     **end**
10     Generate new solution by flying randomly
11     **if** *rand* $> A_i$ and $f(x_i) < f(x_*)$ **then**
12         Accept the new solution
13         Decrease $A_i$, increase $r_i$, by Eqs. 6 and 7
14     **end**
15     Rank the bats and find the current best $x_*$
16 **until** termination criteria is met
17 Post process results and visualization

---

**Figure 3.** Pseudo-code of bat algorithm.

### 2.4. Formulation of the Optimization Problem

The composite optimization model is developed aiming to maximize the net profit of the system, which comes from the sum of the net profit related to the supply chain of manufacturing and logistics. The mathematical formulations of the proposed model are listed in detail below. All the symbols of the variables are also summarized, including the decision variables involved in the optimization process of the proposed model. According to the integrated logistics system structure, the integrated objective function ($\Omega$) of the proposed model contains an objective function whose goal is to maximize the net profit of the manufacturing chain. Therefore, this study constructs a mathematical form of the proposed compound objective function, and the maximum profit is measured by subtracting the corresponding *FC* from the corresponding revenue, given by the following equation:

$$Max\ \Omega = FR - FC = FR - (FPC + FMC + FIC + FTC + FLC + VC) \tag{8}$$

where *FR* = the total amount revenue; *FC* = the total amount of cost; *FPC* = the total amount of cost of acquiring the materials necessary for manufacturing; *FMC* = the total amount of cost of manufacturing; *FIC* = the total cost of all inventory held in stock; *FTC* = the total cost of transportation; *FLC* = the total amount of cost for labor; and *VC* = the total amount of cost of vessel operations. Equation (9) shows the *FC* associated with an existing manufacturing chain is composed of six items which are the corresponding *FPC*, *FMC*, *FIC*, *FTC*, *FLC*, and aggregate operation cost of vessels. The components of the mathematical form shown in Equation (9) are further expressed as presented below:

$$FR = \sum_{\forall t} \left\{ \left[ \sum_{i=1}^{2} \sum_{j=j+1 \forall mi \forall mj}^{3} r_{mi,mj}^{raw}(t) \times W_{mi,mj}(t) \right] + \left[ \sum_{k=2l=4}^{4} \sum_{\forall k}^{5} \sum_{\forall l} r_{mk,ml}(t) \times W_{mk,ml}(t) \right] \right\} \tag{9}$$

where the upper part represents the aggregate profit obtained from the raw fish material flow $W_{mi,mj}(t)$ supplied by the layer of raw material supply to the layer of manufacturing and wholesalers; the part above the formulation introduces the aggregate profit from the physical flow (processes of manufactured product) in any given distribution channel.

$$FPC = \sum_{\forall t} \left\{ \left[ \sum_{i=1}^{2} \sum_{j=i+1}^{3} \sum_{\forall mi} \sum_{\forall mj} c_{mi,mj}^{pro^{raw}}(t) \times W_{mi,mj}(t) \right] + \left[ \sum_{k=2}^{3} \sum_{\forall m4} \sum_{\forall mk} c_{mk,ml}^{pro}(t) \times W_{mk,m4}(t) \right] \right\} \tag{10}$$

where the aggregate cost of obtained material procurement (*FPC*) involves two components: (a) the initial cost of raw materials harvested in the layer of raw material supply and (b) the procurement cost generated from the layer of manufacturing for ordering the raw materials from the raw material supplies of the given manufacturing chain, presented in Equation (10).

$$FMC = \sum_{\forall t} \left\{ \sum_{i=2 \forall mi}^{4} \left[ c_{mi}^{Fz}(t) \times \left( \Psi_{mi}^{inv^{raw}}(t) + \Psi_{mi}^{inv}(t) \right) + c_{mi}^{DEP}(t) + c_{mi}^{MT}(t) \right] + \left[ \sum_{\forall m3} c_{m3}^{man}(t) \times U_{m3}^{man}(t) \right] \right\} \tag{11}$$

where the aggregate manufacturing cost (*FMC*) is composed of three terms: (a) the freeze cost in the layer of raw materials and manufacturing processes in any given distribution channel; (b) the aggregate manufacturing cost for the canned and the processed foods obtained from flows of the manufacturing processes; and (c) the aggregate maintenance

and depreciation costs of equipment in the layer of manufacturing processes, as presented in Equation (11).

$$FIC = \sum_{\forall t} \left\{ \left[ \sum_{i=1}^{3} \sum_{\forall mi} c_{mi}^{inv^{raw}}(t) \times \Psi_{mi}^{inv^{raw}}(t) \right] + \left[ \sum_{i=2}^{4} \sum_{\forall mj} c_{mj}^{inv}(t) \times \Psi_{mj}^{inv}(t) \right] \right\} \quad (12)$$

where the aggregate inventory cost (*FIC*) of the given manufacturing chain involves two components: (a) the inventory cost of the raw materials utilized in both layers of wholesalers and manufacturing processes and (b) the inventory cost of the manufactured products in layers of manufacturing processes and wholesalers in the given manufacturing chain, as presented in Equation (12).

$$FTC = \sum_{\forall t} \left\{ \left[ \sum_{i=1}^{3} \sum_{j=i+1}^{} \sum_{\forall mi} \sum_{\forall mj} c_{mi,mj}^{tp}(t) \times W_{mi,mj}^{tp}(t) \right] + \left[ \sum_{k=2}^{4} \sum_{l=k+1}^{5} \sum_{\forall mk} \sum_{\forall ml} c_{mi,mj}^{tp}(t) \times W_{mi,mj}^{tp}(t) \right] \right\} \quad (13)$$

where the aggregate transportation cost (*FTC*) of the manufacturing chain is composed of the two physical flows: (a) the raw material ($\Psi_{mi,mj}^{tp}(t)$) transported from the layers of raw material to manufacturing plants and (b) the products ($\Psi_{mi,mj}^{tp}(t)$) transported in any given distribution channels of the manufacturing chain, as presented in Equation (13).

$$FLC = \sum_{\forall t} \left\{ \sum_{i=1}^{4} \sum_{\forall mi} c_{mi}^{lab}(t) \times Q_{mi}^{lab}(t) \right\} \quad (14)$$

where the aggregate labor costs (*FLC*) were generated from every process of harvesting, product processing, manufacturing, and in any given distribution channel, as presented in Equation (14).

$$VC = \sum_{i=1}^{3} \frac{(\alpha_i + \beta) \cdot hp_i \cdot hr}{\mathcal{L}} \cdot (1 - \sigma) \cdot \mathcal{KL}_i \quad (15)$$

where the operation costs of vessel (*VC*) are generated from any activities that occurred on the vessel. The determined factor was the fuel that supported power for dynamic. There are three types of fishery vessels in accordance with government regulations. The first type of vessel is supported by pure diesel, the second type of vessel is supported by a mixture of fuel oil and diesel oil, and the last type of vessel is supported by pure fuel oil. The government regulated the standard of the amount of oil purchased that the horsepower of the motor on the fishing vessel corresponded to the tonnage and the corresponding calculation ratio was proposed. For the main and auxiliary engine oil, the corresponding ratios ($\alpha_i + \beta$) are proposed; the amount of oil purchased is calculated by the number of horsepower ($hp_i$) and operating hours ($hr$) corresponding to the tonnage purchased, and the number of liters ($\mathcal{L}$) is a unit converted to kiloliters, and subsidies ($\sigma$) offered by government, in contrast to the published daily oil price ($\mathcal{KL}_i$), are depicted in kiloliters.

### 2.5. Disaggregate Inventory Constraints

Inventory constraints defined the connections of the inbound and outbound logistics flows as well as the corresponding storage quantities associated with the members in the chain.

(1)　For raw material suppliers (mc-layer1),

$$0 \leq W_{m1}^{inv^{raw}}(k) = W_{m1}^{inv^{raw}}(k-1) + W_{m1}^{raw}(k) - \sum_{\forall m2} W_{m1,m2}(k) \leq \Psi_{m1}^{inv^{raw}} \quad \forall(m1,k) \quad (16)$$

There are three parts of raw material supplier equation and include the time-varying inventory amount ($W_{m1}^{inv^{raw}}(k)$) in a given time interval $k$ that is equal to the sum of the cor-

responding inventory remaining amount in the previous time interval $k - 1$ ($W_{m1}^{inv^{raw}}(k-1)$) and the corresponding time-varying amount generated in the given time interval $k$ ($W_{m1}^{raw}(k)$), subtracting the total outbound from raw material flow ($\sum_{\forall m2} Q_{m1,m2}(k)$) transported to the layer of manufacturing in the given time interval $k$. In addition, $W_{m1}^{inv^{raw}}(k)$ is subjected to upper and lower bounds, i.e., the corresponding storage capacity ($\Psi_{m1}^{inv^{raw}}$) and 0. Therefore, Equation (16) is proposed for the disaggregate inventory constraint associated with every raw material supplier ($m1$) in the given supply chain.

(2)  For wholesalers (mc-layers 2),

$$0 \leq W_{m2}^{inv^{raw}}(k) = W_{m2}^{inv^{raw}}(k-1) + \sum_{\forall m1} W_{m1,m2}^{raw}(k) - \sum_{\forall m3} W_{m2,m3}^{inv^{raw}}(k)$$
$$\leq \Psi_{m2}^{inv^{raw}} \quad \forall (m2, k) \tag{17}$$

$$0 \leq W_{m2}^{inv}(k) = W_{m2}^{inv}(k-1) + \sum_{\forall m1} W_{m1,m2}^{raw^{pd}}(k) - \sum_{i=4}^{5}\sum_{\forall mi} W_{m2,mi}^{inv}(k)$$
$$\leq \Psi_{m2}^{inv} \quad \forall (m2, k) \tag{18}$$

The disaggregate inventory ($W_{m2}^{inv}(k)$) was varied with time, and amounts were constrained by upper and lower bound with given wholesalers. There was a difference between these two types of inventories. When raw materials ($W_{m1,m2}^{raw}(k)$) moved to the layer of wholesaler, a part of raw materials would be transformed to products ($W_{m1,m2}^{raw^{pd}}(k)$) for selling directly to retailers and customers via processing of repackaging, as presented in Equation (18). Then, the other part of raw materials ($W_{m1,m2}^{raw}(k)$) would be similarly transported to the manufacturer as raw materials, as presented in Equation (17).

(3)  For product manufacturers (mc-layer 3),

$$0 \leq W_{m3}^{inv^{raw}}(k) = W_{m3}^{inv^{raw}}(k-1) + \sum_{i=1}^{2}\sum_{\forall m3} W_{mi,m3}^{inv^{raw}}(k) - W_{m3}^{man}(k) \leq \Psi_{m3}^{inv^{raw}} \quad \forall (m3, k) \tag{19}$$

$$0 \leq W_{m3}^{inv}(k) = W_{m3}^{inv}(k-1) + W_{m3}^{man}(k) - \sum_{i=4}^{5}\sum_{\forall mi} W_{m3,mi}^{inv}(k)$$
$$\leq \Psi_{m3}^{inv} \quad \forall (m3, k) \tag{20}$$

The amounts of time-varying inventory can be divided into two parts: raw materials ($W_{m3}^{inv^{raw}}(k)$) and manufacturer products ($W_{m3}^{inv}(k)$), and the rationales are applied in Equations (17) and (18). The variation of channel distribution transformed from raw materials to products should be considered. The raw materials ($W_{mi,m3}^{inv^{raw}}(k)$) from raw material suppliers and wholesalers could have been processed as fishery products ($W_{m3}^{man}(k)$), the finished products can be transferred to retailers and customers as presented in Equations (19) and (20).

(4)  For retailers (mc-layer 4),

$$0 \leq W_{m4}^{inv}(k) = W_{m4}^{inv}(k-1) + \sum_{i=2}^{3}\sum_{\forall mi} W_{mi,m4}^{inv}(k) - W_{m4,m5}^{inv}(k)$$
$$\leq \Psi_{m4}^{inv} \quad \forall (m4, k) \tag{21}$$

Based on the same rationale, the time-varying inventory ($W_{m4}^{inv}(k)$) is subjected to the upper and lower bounds with the amounts of inventory ($\Psi_{m4}^{inv}$) with any given retailers in the chain. There were three parts shown in this equation (Equation (21)); the first part defined the remaining inventory ($W_{m4}^{inv}(k-1)$); the second part presented products obtained from wholesalers and manufacturers transported to the layer of retailers; and the last part presented the product transferred to the customers.

## 3. Experimental and Numerical Results

This section presents the experimental analysis and design conducted using the BA to solve the optimization problem in the perishable product industry. To illustrate the applicability of the method, a simplified numerical study was implemented. This study constructed an integrated logistic network based on fishery activities in the north of Taiwan including the logistics distribution channels. After extracting related information, the estimated data input such as parameters and constraints of the model, and the core parameters, cost, and revenue involved in the whole processes in the supply chain were used to formulate corresponding supply chain management issues. The numerical results of the optimal solution were determined by using the algorithm, which was then compared with the operating performance of the optimized approach. The significant procedures of the numerical study and the thorough details of the corresponding results are presented in this section.

### 3.1. Parameter Setting

The estimation of relevant parameters was acquired through the interview data. Due to the difficulty in obtaining some confidential government information, it is hard to estimate related parameters directly from the statistical data of the information, such as unit operating costs and revenue. To tackle such a difficulty, this study incorporates the Fisheries Agency and the Department of Environmental Biology and Fishery Sciences of National Taiwan Ocean University for assistance by interviewing the experts. This study carried out one round of interviews using a purposeful sampling technique of professors from National Taiwan Ocean University. The representation and reliability of experts were based on their extensive years of research experience (a minimum of 10 years) in supply chain management and logistics and the number of published research papers in SSCI/SCI journals. The experts were approached in person, and thus, the interviews were conducted in each of the expert's office rooms. Each interview took up to one hour. The first 15 min before the interview was focused on introducing and explaining the research. The rest 45 min were concentrated on identifying the relevant parameters needed for the research. Overall, the collection timespan was three weeks due to the different availability of the experts. Thus, this study uses a qualitative method by employing a complex cost function with multiple restrictions to define the relation and analysis sensitivity of quantity and price. The interview was focused on the potential operational performance (e.g., the probable range of operating revenues and costs) and the corresponding logistic constraints involving cold chain technology such as the capacity of the facility, availability of fleet size, and frequency of vehicle scheduling. The collected data through the interview were analyzed and processed to generate the corresponding upper and lower bounds of cost and revenue.

The information was analyzed using a uniform distribution, and the ranges were respectively defined by the estimated upper and lower bounds. Then, the corresponding unit revenues and costs were associated with each chain member in the proposed model. The survey data herein was used to set limits on the corresponding parameters with the upper and lower bounds by uniform distribution functions. The estimated intervals of the corresponding revenue are summarized in Table 3, whereas the corresponding unit ranges and costs involved in the manufacturing supply chain are summarized in Table 4. The other main parameters were prearranged depending on the data previously mentioned, as shown in Table 5.

**Table 3.** Estimated boundaries of unit revenues.

| Layer-Mc | Parameter | Unit Revenues (kg/US) | |
| --- | --- | --- | --- |
| | | Lower Bound | Upper Bound |
| $m1 \rightarrow$ (raw materials) $\rightarrow m2$ | $r^{raw}_{m1,m2}(t)$ | 2.74 | 4.79 |
| $m1 \rightarrow$ (raw materials) $\rightarrow m3$ | $r^{raw}_{m1,m3}(t)$ | 2.74 | 4.79 |
| $m2 \rightarrow$ (raw materials) $\rightarrow m3$ | $r^{raw}_{m2,m3}(t)$ | 1.35 | 2.33 |
| $m2 \rightarrow$ (products) $\rightarrow m4$ | $r_{m2,m4}(t)$ | 2.39 | 5.17 |
| $m2 \rightarrow$ (products) $\rightarrow m5$ | $r_{m2,m5}(t)$ | 4.24 | 5.46 |
| $m3 \rightarrow$ (products) $\rightarrow m4$ | $r_{m3,m4}(t)$ | 1.94 | 3.35 |
| $m3 \rightarrow$ (products) $\rightarrow m5$ | $r_{m3,m5}(t)$ | 2.43 | 4.2 |
| $m4 \rightarrow$ (products) $\rightarrow m5$ | $r_{m4,m5}(t)$ | 4.26 | 6.47 |

**Table 4.** Estimated boundaries of unit costs associated with the manufacturing supply chain.

| Layer-Mc | Parameter | Unit Revenues (kg/US) | |
| --- | --- | --- | --- |
| | | Lower Bound | Upper Bound |
| $m1$: raw material suppliers | $c^{inv^{raw}}_{m1}(t)$ | 0.3 | 0.5 |
| | $c^{lab}_{m1}(t)$ | -- | 30 |
| $m2$: wholesalers | $c^{pro^{raw}}_{m1,m2}(t)$ | 2.37 | 4.16 |
| | $c^{Fz}_{m2}(t)$ | 0.88 | 1.12 |
| | $c^{DEP}_{m2}(t)$ | -- | 100 |
| | $c^{MT}_{m2}(t)$ | -- | 250 |
| | $c^{inv^{raw}}_{m2}(t)$ | 0.3 | 0.5 |
| | $c^{inv}_{m2}(t)$ | 0.3 | 0.5 |
| | $c^{tp}_{m1,m2}(t)$ | 0.25 | 0.5 |
| | $c^{lab}_{m2}(t)$ | -- | 25 |
| $m3$: manufacturer | $c^{pro^{raw}}_{m1,m3}(t)$ | 2.38 | 4.16 |
| | $c^{pro^{raw}}_{m2,m3}(t)$ | 1.35 | 2.33 |
| | $c^{man}_{m3}(t)$ | -- | 0.94 |
| | $c^{Fz}_{m3}(t)$ | 0.88 | 1.12 |
| | $c^{DEP}_{m3}(t)$ | -- | 1000 |
| | $c^{MT}_{m3}(t)$ | -- | 500 |
| | $c^{inv^{raw}}_{m3}(t)$ | 0.3 | 0.5 |
| | $c^{inv}_{m3}(t)$ | 0.38 | 0.61 |
| | $c^{tp}_{m1,m3}(t)$ | 0.25 | 0.5 |
| | $c^{tp}_{m2,m3}(t)$ | 0.25 | 0.5 |
| | $c^{lab}_{m2}(t)$ | -- | 30 |
| $m4$: retailers | $c^{pro}_{m2,m4}(t)$ | 2.72 | 4.14 |
| | $c^{pro}_{m3,m4}(t)$ | 2.74 | 4.79 |
| | $c^{Fz}_{m4}(t)$ | 0.88 | 1.12 |
| | $c^{DEP}_{m4}(t)$ | -- | 100 |
| | $c^{MT}_{m4}(t)$ | -- | 100 |
| | $c^{inv}_{m4}(t)$ | 0.5 | 1.00 |
| | $c^{tp}_{m2,m4}(t)$ | 0.35 | 0.6 |
| | $c^{tp}_{m3,m4}(t)$ | 0.35 | 0.6 |
| | $c^{lab}_{m4}(t)$ | -- | 15 |
| $m5$: customers | $c^{tp}_{m2,m5}(t)$ | 0.35 | 0.6 |
| | $c^{tp}_{m3,m5}(t)$ | 0.35 | 0.6 |
| | $c^{tp}_{m5,m5}(t)$ | 0.35 | 0.6 |

**Table 5.** Summary of primary related parameter.

| | | | | | |
|---|---|---|---|---|---|
| 1. | Inventory capacity | | | | |
| $\Psi_{m1}^{inv^{raw}}$ | $\Psi_{m2}^{inv^{raw}}$ | $\Psi_{m2}^{inv}$ | $\Psi_{m3}^{inv^{raw}}$ | $\Psi_{m3}^{inv}$ | $\Psi_{m4}^{inv}$ |
| 5000 | 2000 | 1000 | 2000 | 3000 | 1000 |
| 2. | Corresponding gas rate | | | | |
| $\alpha_1$ | $\alpha_2$ | $\alpha_3$ | $\beta$ | $\sigma$ | |
| 0.18 | 0.3 | 0.18 | 0.18 | 0.14 | |
| 3. | Corresponding rate of time and hour | $hr^1 = 72$ | | | |
| 4. | Corresponding capacity of gas | | | | |
| $\mathcal{L}$ | $\mathcal{KL}_1$ | $\mathcal{KL}_2$ | $\mathcal{KL}_3$ | | |
| 1000 | 596.43 | 466.23 | 589.77 | | |

[1] in accordance with governmental regulation, the maximum fuel consumption was limited to 72 h.

### 3.2. Analysis of Numerical Results

The programming approach in this study was processed to generate the result through Python with modified code. This study evaluated the performance of the proposed method by comparing the result and efficiency with nonlinear programming and the BA, respectively, using the given data and parameters. The nonlinear programming and the BA were implemented by using the programming language, Python. Despite using the same programming language, the former used the approach of optimizing solutions, and the latter adopted the approach of the BA in Python.

Accordingly, this study has obtained the arithmetic results as summarized in Table 6. The optimized solution was compared with the one generated by the BA, which leans toward the optimized solution. Moreover, the results from the BA may change as the fitness generated randomly would not be closest to the best solution during the iterations in every execution. The starting positions and movement of the bats are different at each time of execution, and in this situation, this study proposes a comparison of the BA and optimized solution, and the comparison of the best BA and the optimized solution is shown in Table 6.

**Table 6.** Performance evaluation of approach using the proposed model.

| Evaluation Criterion<br>Operation Alternative | Net Profit (USD) |
|---|---|
| Non-linear programming | 41,850 |
| Bat Algorithm approach (80 bats) | 41,515 |

### 3.3. Sensitivity Analysis of Cost Allocation

Cost is crucial to the aggregate profit that directly affects the structure between revenue and cost. For suppliers to understand how the costs vary with overall profit, it was analyzed in a commodity-oriented manner, and the commodity types were divided into raw materials and processed products in this section. The sensitivity analysis was employed with adjustment to parameters, and the inventory limits of manufacturers and wholesalers were varied to observe the effects on aggregate profit. The corresponding results were listed in Scene 1 and Scene 2, as can be seen in Table 7.

**Table 7.** Performance evaluation of approach with varying inventory limits of wholesalers and manufacturers.

| Evaluation Criterion<br>Operation Alternative | Net Profit (USD) |
|---|---|
| Scene 1 (Raise inventory of $m2$) | 66,290 |
| Scene 2 (Raise inventory of $m3$) | 68,240 |
| The optimal method | 41,515 |

The results show that in Scene 1 with a raised inventory limit of wholesalers, the net profit is USD 66,290, while in Scene 2 with a raised inventory limit of manufacturers, the net profit is USD 68,240. Table 8 shows the comparison of both scenes based on revenues and costs to identify the profit.

**Table 8.** Composition of profit with varying inventory limits of wholesalers and manufacturers.

| | Scene 1 Raise Inventory of $m2$ (USD) | | Scene 2 Raise Inventory of $m3$ (USD) | |
|---|---|---|---|---|
| Aggregate revenue | 157,940 | | 168,230 | |
| Aggregate cost | | | | |
| Procurement cost | | 34,550 | | 35,940 |
| Manufactured cost | | 22,220 | | 26,340 |
| Inventory cost | | 14,830 | | 16,160 |
| Transportation cost | | 16,300 | | 17,800 |
| Aggregate profit | 66,290 | | 68,240 | |

## 4. Discussion

This section presents a discussion of the results divided into theoretical and managerial implications.

### 4.1. Theoretical Implication

Based on the results in Table 8, there is a difference in aggregate profit when wholesalers ($m2$) and manufacturers ($m3$) are compared. The results imply that aggregate profit is optimized by raising the inventory of the manufacturers ($m3$) to USD 68,240 in comparison to raising that of the wholesalers ($m2$) to USD 66,290. This implies that among the involved players—raw material suppliers, wholesalers, retailers, consumers, and manufacturers—that influence the overall supply chain performance, the manufacturers play a significant role in improving the model in terms of yielding optimal profits. In general, the recognized relationship between manufacturers and the overall supply chain has been discussed with a positive tone in other studies where manufacturing firms can use supply chain practices to improve supply chain collaboration and performance, and a manufacturer's degree of relational embeddedness is a crucial factor in its decision to establish a direct connection between its suppliers and customers. Additionally, the manufacturer's reported ability to steer and manage this relationship underscores its role as a central player in orchestrating interactions within the supply chain network [14,15]. The manufacturer's role in the supply chain is emphasized in product packing, cold-storage, labor cost, transportation and deterioration, and maintenance [26]. However, this study suggests that the roles of the other stakeholders are not to be underestimated. For instance, there is a sensitive line between the role of manufacturers and that of retailers that effectively disturbs the manufacturers' financial gain subject to aspects such as the retailer's product rebranding, profitability analysis, and profit-oriented strategies [17]. Nevertheless, this study highlights the importance of strengthening the manufacturers' role throughout the supply chain processes to enhance the chain's overall performance.

### 4.2. Managerial Implication

Overall, the findings show that each parameter significantly impacts the numerical outcome, which may then be used to inform prospective managerial suggestions for Taiwan's relevant government institutions or agencies that have expertise in dealing with the fishing industry's supply chain. However, the inventory limit is a critical determinant affecting the performance of the model in this study. Specifically, the results reveal that profit maximization is sensitively influenced by the changes in setting the inventory limit, which interacts with the revenue and cost. After raising the inventory limit of both wholesalers and manufacturer, the results show that the rise in the inventory limit of the manufacturer yields a higher profit. In other words, by raising the inventory limit of the

manufacturer from the original quantity to twice, the structure of aggregate profit becomes observable. Thus, in practice, this study suggests decision-makers at the manufacturer's end perform proper inventory management so that consumers can place their orders with confidence, by optimizing the stock levels and minimizing unused or perished fishery products; hence, the sales increase to boost the profits to the maximum levels, which allows costs to be reduced and revenue to be increased. The corresponding numerical results indicate that the profit in this model potentially increases significantly with an increasing quantity of processed products from the manufacturer. Therefore, this study suggests that decision-makers concentrate on allocating quantity of fisheries while taking into account how revenue and cost impact the model. Nevertheless, the model is adjustable in response to different conditions.

### 4.3. Political and Academic Implications

Based on the findings, the political implications can be highlighted on the policy formulation and industry regulation through (1) the potential to influence policy formulation and regulations related to the fishing industry's supply chain. Specifically, the identification of manufacturers' significant role in optimizing profits suggests that policymakers and relevant government institutions need to consider targeted strategies and incentives to support manufacturers within the supply chain; (2) inter-stakeholder collaborations, including raw material suppliers, wholesalers, retailers, consumers, and manufacturers; and (3) sustainable resource management within the fishing industry's supply chain to prevent overfishing, reduce waste, and promote responsible sourcing.

Furthermore, this study offers several academic implications by (1) emphasizing the advancement of supply chain theory through a comprehensive understanding of each stakeholder's role in achieving optimal outcomes; (2) contrasting prior studies to critically analyze the contextual factors that shape differing perspectives on manufacturers' contributions to supply chain performance; and (3) model flexibility and applicability might be explored by the academics to be tailored to different industry contexts, considering factors like revenue, cost, and stakeholder interactions.

### 5. Conclusions

Due to the dominance of Taiwan's maritime location, both deep-sea fishery and offshore fishery sectors have prospered. However, the issue of offshore fisheries is rarely researched due to the complexity arising from fish species and harvest seasons, which significantly complicates the proposed profit model structure in this study. The primary objective of this research is to develop an optimized model specifically tailored for offshore fisheries. The study takes into account the basic yet essential conditions to facilitate future research expansion. An integrated logistics model has been proposed to coordinate the flow of raw materials and processed products within the logistics framework of the offshore fishery environment in Taiwan. By determining the key factors and associated operational conditions in the proposed integrated logistics system, a comprehensive objective function and corresponding constraints have been formulated. The recently proposed heuristic algorithm, BA, known for its efficiency and adaptability across various problem types, is employed in this study. The results obtained using the BA show a slight deviation from the best solution, providing evidence of its proximity to optimality. The numerical results of the sensitivity analysis, focusing on raising the inventory limit, indicate that a higher inventory limit for manufacturers can generate increased profits, enabling suppliers to supply fishery materials to more lucrative trading partners. In this study, the profit performance of manufacturers and wholesalers plays a crucial role. Furthermore, this research is based on a small-scale and basic revenue and cost model, but the objective and constraints would become more complex when considering varying circumstances in the logistic system. Future research can utilize this model to expand its scale and incorporate practical cases for further development. Moreover, the model is limited to several key players and can be extended by involving the role of distributor and fish collectors from the fishermen. In

addition, the model is examined based on profit maximization with two examined scenes involving the role of wholesalers and manufacturers in raising the inventory limit. Future studies can examine more scenes by involving more players and decision-makers in the scenarios. Moreover, the future study should consider other aspects that are more crucial to develop sustainability rather than focusing on the profit maximization as a sole goal. At last, this study is focused on the offshore fishery industry in Taiwan which is unique to specific characteristics due to the geographic location of the country. Thus, future studies might make a model comparison of the different regions with their unique characteristics.

**Author Contributions:** Conceptualization, formal analysis, writing—original draft preparation, software, validation, M.-F.Y.; conceptualization, investigation, resources supervision, S.-L.K.; conceptualization, writing—original draft preparation, writing – review and editing, project administration, R.Y.S. All authors have read and agreed to the published version of the manuscript.

**Funding:** This research received no external funding.

**Institutional Review Board Statement:** Not applicable.

**Informed Consent Statement:** Not applicable.

**Data Availability Statement:** The data presented in this study are available on request from the corresponding author.

**Conflicts of Interest:** The authors declare no conflict of interest.

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
