# Peer review of "The Development of the Optimal Harvesting Model of an Offshore Fishery Supply Chain Based on a Single Vessel"

_jmse, doi:10.3390/jmse11081593_

Round 1

Reviewer 1 Report (Previous Reviewer 2)

No further comments

Author Response

We sincerely appreciate your time and expertise in reviewing our manuscript. 

Reviewer 2 Report (Previous Reviewer 3)

On 16-5-23, I reviewed your previous version of this paper (JMSE 24132422) as follows:

“This paper is a study of the supply chain of Taiwan’s offshore fishery. You evidently seek to make this supply chain more efficient and examine many links in it with a view to improving its performance. I am not qualified to comment on the mathematical modelling methods you employed, so another reviewer will have to evaluate them. My own comments are five-fold.   

(1)    You do not tell us why you are interested in this topic. Are you implying that the existing supply chain is inefficient? If so, you should explain what these inefficiencies are. In the Abstract, you say “there are challenges that hinder the offshore fishery supply chain” (lines 10-11). But you don’t explain what these challenges are or how they currently hinder the supply chain. In lines 32-34, you say “wild fishing depends on fish species, weather conditions and government regulations. These conditions have caused challenges in the offshore fishery supply chain”. In lines 498-500, you say “the fishery industry faces several challenges due to fish species availability in different seasons, weather conditions and relevant regulations, which affect the fishery supply chain”. But again, you don’t explain what these challenges are or how they affect the supply chain. Moreover, you say Taiwan’s offshore fishery is prosperous:

The Taiwanese offshore fishing industry has grown prosperously in recent years. The aquatic product market has turned export-oriented, facilitating the product supply chain to develop more maturely along with the cold chain technology development” (lines 26-28).

“Due to the abundance of the marine resources, the deep-sea fishery and offshore fishery are both prosperous in Taiwan” (lines 496-497

If it ain’t broke, why fix it?

(2)    You assert that “it is necessary to develop an optimal harvesting model under the offshore fishery supply chain evaluated based on profit maximization” (lines 13-15). But why is it necessary to construct a model based on profit maximisation? For many researchers profit maximisation is not necessarily a desirable objective, especially if it is at the expense of other values, You need to defend your assumption that profit maximisation is the one and only objective.  

(3)    You refer to interview data:

The estimation of relevant parameters was acquired through the interview data. Due to a difficulty in obtaining some confidential government information, it is hard to estimate related paramaters directly from the statistical data of the information, such as unit operating costs and revenue. To tackle such a difficulty, this study incorporates the Fisheries Agency and the Department of Environmental Biology and Fishery Sciences of National Taiwan Ocean University for assistance by interviewing the experts. The interview was focused on the potential operational performance (e.g., the probable range of operating revenues and costs) and the corresponding logistic constraints involving cold chain technology such as the capacity of facility, availability of fleet size, and frequency on vehicle scheduling. The collected data through the interviews were analyzed and processed to generate the corresponding upper and lower bounds of cost and revenue” (lines 394-404).

But you do not explain how many interviews were carried out; how the interviewees were selected; how representative were the interviewees; nor how reliable were their responses. Moreover, if you are here making use of qualitative methods in a paper which is otherwise based on quantitative methodology, you should explain and justify your mixed methods.    

(4)    Some of your ‘findings’ are banal. For instance, on lines 460-463, you say:

“The results in Table 3.7 indicate that among the involved players – raw material suppliers, wholesalers, retailers, consumers, manufacturers – that influence the overall supply chain performance, the manufacturers play a significant role in improving the model in terms of yielding the optimal profits”.

And on lines 508-510, you say:

“The findings indicate that among the key players throughout the supply chain processes, manufacturers play a crucial policy- and decision-making role in enhancing the chain performance”

               Surely it is blindingly obvious that manufacturers play a crucial role in supply chain performance?  

(5)    Generally well-written, your paper has some infelicities of expression and some spelling errors. For example, the following passage on lines 155-159 is so dense with jargon that it is well-nigh unintelligible, and it has also misspelled two words:   

“Moreover, a hybrid model was proposed using Lagrangian Relaxation (LR) method, meta-heuristic method, bat algorithm and practicle swarm optimization method to solve the long-term production scheduling problem caused by determinisitc assumptions and level uncertainties. Such a hybrid provides a near-optimal solution”.

My review of your resubmitted version (JMSE 2525952) is that you have addressed my first and fifth comments, but not my second, third or fourth comments. For example, on my second comment, you do not justify your assumption that profit maximisation is the only goal. On my third comment, you have not explained or justified your interview protocol. On my fourth comment, you have not addressed my criticism that some of your conclusions are banal. Accordingly, I have no option but to reject your new submission. 

Author Response

Reviewer #2

Authors response

My review of your resubmitted version (JMSE 2525952) is that you have addressed my first and fifth comments, but not my second, third or fourth comments. For example, on my second comment, you do not justify your assumption that profit maximisation is the only goal. On my third comment, you have not explained or justified your interview protocol. On my fourth comment, you have not addressed my criticism that some of your conclusions are banal. Accordingly, I have no option but to reject your new submission. 

Thank you for the feedback.

We highly appreciate the thorough review of the resubmitted version and specific feedback regarding the addressed and unresolved comments. We understand the concerns and will provide further clarification on the points in the following revision.

(2)    You assert that “it is necessary to develop an optimal harvesting model under the offshore fishery supply chain evaluated based on profit maximization” (lines 13-15). But why is it necessary to construct a model based on profit maximisation? For many researchers profit maximisation is not necessarily a desirable objective, especially if it is at the expense of other values, You need to defend your assumption that profit maximisation is the one and only objective.  

Thank you for the fruitful comment.

We appreciate your critical examination of the assertion regarding the necessity of developing an optimal harvesting model based on profit maximization.

In this study, we focus on the profit maximization model using the parameters that are commonly used in prior studies that focused on the profit maximization model. However, we share your concern regarding the importance to recognize that while profit maximization can be a significant consideration in business and economic contexts, it might not always align with broader societal, ethical, and environmental goals. As a result, we view your criticism as a limitation to this study which will be addressed in a follow-up study using a more comprehensive and inclusive approach that considers ecological, social, and ethical dimensions, which are essential for creating a sustainable and balanced outcome that benefits all stakeholders involved.

In this study, constructing a profit-maximizing model in the offshore fishery industry is necessary for optimizing the supply chain, meeting market demand, managing resources, and addressing complex supply chain problems. Furthermore, it assesses the impacts of industry development on cost problems and fishery management, improving coordination and efficiency. Overall, the profit-maximizing model enables efficient supply chain management in the offshore fishery industry.

(3)    You refer to interview data:

The estimation of relevant parameters was acquired through the interview data. Due to a difficulty in obtaining some confidential government information, it is hard to estimate related paramaters directly from the statistical data of the information, such as unit operating costs and revenue. To tackle such a difficulty, this study incorporates the Fisheries Agency and the Department of Environmental Biology and Fishery Sciences of National Taiwan Ocean University for assistance by interviewing the experts. The interview was focused on the potential operational performance (e.g., the probable range of operating revenues and costs) and the corresponding logistic constraints involving cold chain technology such as the capacity of facility, availability of fleet size, and frequency on vehicle scheduling. The collected data through the interviews were analyzed and processed to generate the corresponding upper and lower bounds of cost and revenue” (lines 394-404).

But you do not explain how many interviews were carried out; how the interviewees were selected; how representative were the interviewees; nor how reliable were their responses.

Moreover, if you are here making use of qualitative methods in a paper which is otherwise based on quantitative methodology, you should explain and justify your mixed methods.

Thank you for the fruitful comment.

We have addressed the concern regarding the interview protocol in the following revision.

“The estimation of relevant parameters was acquired through the interview data. Due to the difficulty in obtaining some confidential government information, it is hard to estimate related parameters directly from the statistical data of the information, such as unit operating costs and revenue. To tackle such a difficulty, this study incorporates the Fisheries Agency and the Department of Environmental Biology and Fishery Sciences of National Taiwan Ocean University for assistance by interviewing the experts. This study carried out one round of interviews using a purposive sampling technique of professors from National Taiwan Ocean University. The representation and reliability of experts were based on their extensive years of research experience (at a minimum of 10 years) in supply chain management and logistics and the number of published research papers in SSCI/SCI journals. The experts were approached in person, and thus the interviews were conducted in each of the expert’s office rooms. Each interview took up to one hour. The first 15 minutes before the interview was focused on introducing and explaining the research. The rest 45 minutes were concentrated on identifying the relevant parameters needed for the research. Overall, the collection timespan was three weeks due to the different availability of the experts. Thus, this study uses a qualitative method by employing a complex cost function with multiple restrictions to define the relation and analysis sensitivity of quantity and price. The interview was focused on the potential operational performance (e.g., the probable range of operating revenues and costs) and the corresponding logistic constraints involving cold chain technology such as the capacity of the facility, availability of fleet size, and frequency of vehicle scheduling. The collected data through the interview were analyzed and processed to generate the corresponding upper and lower bounds of cost and revenue.”

(4)    Some of your ‘findings’ are banal. For instance, on lines 460-463, you say:

“The results in Table 3.7 indicate that among the involved players – raw material suppliers, wholesalers, retailers, consumers, manufacturers – that influence the overall supply chain performance, the manufacturers play a significant role in improving the model in terms of yielding the optimal profits”.

And on lines 508-510, you say:

“The findings indicate that among the key players throughout the supply chain processes, manufacturers play a crucial policy- and decision-making role in enhancing the chain performance”

Surely it is blindingly obvious that manufacturers play a crucial role in supply chain performance?

Thank for you the fruitful comment.

We appreciate your feedback and understand your concern regarding the perceived banality of the findings related to the role of manufacturers in the supply chain performance. It is indeed evident to many that manufacturers play a crucial role in the supply chain process. However, the purpose of the mentioned statements in this study is to reinforce the importance of manufacturers within the context of the research. In other words, the finding serves as a reminder of the fundamental role that manufacturers play in supply chain performance, particularly within the framework of this study.

Moreover, the findings might be useful for the readers who may not be familiar with supply chain dynamics. Therefore, restating the significant role of manufacturers in supply chain can be helpful for readers who are new to the topic or seeking a comprehensive understanding of the research's implications.

In summary, despite the apparent significance of manufacturers in the supply chain performance, the manufacturers still play a vital role in contextualizing the research and contributing to the existing knowledge on the subject. Through this study, the findings are also to make alignment with previous work in the field.

Reviewer 3 Report (New Reviewer)

Dear Authors,
This was a nice paper to read, and I believe the topic covered can also arouse interest in places other than Taiwan.
The article is well-written, and the limits of the study and its future research are well-explained.
Even the methodological part is well written and easy to understand, even by someone like me who is not an expert on the algorithm used.
Consequently, my comments primarily concern the form and not the document's substance.
Being an article without real case studies, the part dedicated to the literature review is expected to be a lot. Still, I think it is excessive and loses the discussion thread.
For example, in the first reading, I asked myself why doing both paragraphs 2.3 (chapter 2) and 2.3 (chapter 3, please solve the mistake).
I suggest bundling the review literature with the introduction and put in appendix tables 1 and 2.
Regarding the discussion, I feel it necessary to offer it more space to highlight the contributions and food for thought that this article can offer.
For example, I would have liked more comments on what you wrote in lines 560-562.
Furthermore, although they may be few, each article has implications for politics and academia. I, therefore, suggest that these aspects be made more explicit.

Best regards

Author Response

Reviewer #3

Authors response

Dear Authors,
This was a nice paper to read, and I believe the topic covered can also arouse interest in places other than Taiwan.
The article is well-written, and the limits of the study and its future research are well-explained.
Even the methodological part is well written and easy to understand, even by someone like me who is not an expert on the algorithm used.
Consequently, my comments primarily concern the form and not the document's substance. 

Thank you for the feedback.

We highly appreciate your feedback and comments on our study.  

Being an article without real case studies, the part dedicated to the literature review is expected to be a lot. Still, I think it is excessive and loses the discussion thread.

For example, in the first reading, I asked myself why doing both paragraphs 2.3 (chapter 2) and 2.3 (chapter 3, please solve the mistake).

Thank you for the fruitful comment.

We have rearranged the chapters according to the suggestion.

I suggest bundling the review literature with the introduction and put in appendix tables 1 and 2.

Thank you for the suggestion.

We have rearranged the chapters by bundling the literature review with the introduction and moved Tables 1 and 2 to the appendix, according to the suggestion.  

Regarding the discussion, I feel it necessary to offer it more space to highlight the contributions and food for thought that this article can offer.

For example, I would have liked more comments on what you wrote in lines 560-562.

Furthermore, although they may be few, each article has implications for politics and academia. I, therefore, suggest that these aspects be made more explicit.

Thank you for the suggestion.

We have added more explanation to the addressed lines, as follows.

“In general, the recognized relationship between manufacturers and the overall supply chain has been discussed with a positive tone in other studies where manufacturing firms can use supply chain practices to improve supply chain collaboration and performance, and that a manufacturer's degree of relational embeddedness is a crucial factor in its decision to establish a direct connection between its suppliers and customers. Additionally, the manufacturer's reported ability to steer and manage this relationship underscores its role as a central player in orchestrating interactions within the supply chain network [14,15].”

We have also highlighted the political and academic implications, as follows.

“4.3 Political and academic implications

Based on the findings, the political implications can be highlighted on the policy formulation and industry regulation through (1) the potential to influence policy formulation and regulations related to the fishing industry’s supply chain. Specifically, the identification of manufacturers’ significant role in optimizing profits suggests that policymakers and relevant government institutions need to consider targeted strategies and incentives to support manufacturers within the supply chain; (2) inter-stakeholder collaboration, including raw material suppliers, wholesalers, retailers, consumers, and manufacturers; and, (3) sustainable resource management within the fishing industry’s supply chain to prevent overfishing, reduce waste, and promote responsible sourcing.

Furthermore, this study offers several academic implications by (1) emphasizing the advancement of supply chain theory through a comprehensive understanding of each stakeholder’s role in achieving optimal outcomes; (2) contrasting prior studies to critically analyze the contextual factors that shape differing perspectives on manufacturers’ contributions to supply chain performance; and (3) model flexibility and applicability might be explored by the academics to be tailored to different industry contexts, considering factors like revenue, cost, and stakeholder interactions.”

Round 2

Reviewer 2 Report (Previous Reviewer 3)

You have addressed my residual queries adequately so I have recommended publication of your revised MSS. 

This manuscript is a resubmission of an earlier submission. The following is a list of the peer review reports and author responses from that submission.

Round 1

Reviewer 1 Report

jmse-2413422-peer-review-v1
Optimizing supply chain model in Taiwan’s offshore fishery: Manufacturer’s role in raising inventory limit.

In this paper, author shares a study on the optimizing supply chain model in Taiwan’s offshore fishery and manufacturer’s role in raising inventory limit. Although the subject is interesting, I cannot accept the paper due to some serious technical reasons. Contribution seen in the paper is very less.  Section 3, 4, 5 are elementary. There are many technical flaws in this paper. Also in introduction there is no motivation and research gaps seen.

I feel that although this may be a useful exercise, the paper does not come up to the international standards for publication. There is hardly any original contribution in this paper and it adds very little to the published literature.

Some comments may be useful to author:

1.             The title is not clear, appealing, interesting and specific. I suggest to revise the paper title to make it more concise and suitable.

2.             In abstract, some sentences are lengthy and unclear. The entire abstract should be in present tense. The author is suggested to revise all lengthy sentences available in abstract as well as in entire paper.

3.             Lastly, I also notice some references contain errors, missing or incorrect information, and inconsistent formatting. The authors should correct this with the best care.

***

Extensive editing of English language required

Author Response

Ms. Ref. No.: jmse-2413422

Dear Editors and Reviewers

Greetings

We thank you for the generous comments, feedback, and suggestions on the manuscript ID: jmse-2413422 for the improvement of our study.

We have specifically answered the comments from the reviewers in the tables below.

Best regards,

The authors

________________________

Reviewer comments

Reviewer #1

Authors response

The title is not clear, appealing, interesting and specific. I suggest to revise the paper title to make it more concise and suitable.

Thank you for the suggestion.

The authors changed the paper title to “The development of the optimal harvesting model of an offshore fishery supply chain based on single vessel”

   In abstract, some sentences are lengthy and unclear. The entire abstract should be in present tense. The author is suggested to revise all lengthy sentences available in abstract as well as in entire paper.

Thank you for the suggestion.

The abstract has been revised to be less lengthy and clear, as follows.

“This study aims to optimize a model of an offshore fishery supply chain based on a single vessel. Prior studies tend to concentrate on exploring the economic output value of a single fish species, while the offshore fishery supply chain remains understudied. However, some challenges hinder the offshore fishery supply chain. A supply chain model is proposed by incorporating a goal formula that maximizes profits and a restrictive formula based on the conditions of the offshore fishery environment, the fishing equipment used by most fishermen, and other related conditions. A profit maximization model and a restrictive formula based on the offshore fishery environment are established. In addition, the bat algorithm and non-linear programming are applied to develop the solution algorithm, and Python is used to solve the method’s problem. According to the results, manufacturers are crucial to the supply chain's ability to maximize profitability. Moreover, among all the considered parameters, this study concentrates the managerial recommendation on the parameter of inventory limit as a critical determinant to enhance the supply chain performance while generating the highest profits.”

Lastly, I also notice some references contain errors, missing or incorrect information, and inconsistent formatting. The authors should correct this with the best care.

Thank you for the fruitful comment.

The authors have cross-checked the references.

Reviewer 2 Report

 Interesting adaptation of Bat algorithm for optimization problems in offshore fishery suplly chain.

Several doubt or suggestions:

Figure 1: unclear: what elements in this diagram show that it is about integrated logistics? Or maybe it is rather an illustration of the assumptions of flows of goods and reverse flows of funds between participantsRow 113: manufactured plants or manufacturing plants? Care on accuracy of description.

Rows 175-176: optimal model or model used for optimization? As it is stated in the introductory part, the problem of optimization in supply chain is considered.

Rows 198-202: There is no information about keyword appropriate, according to objective of the paper such as: supply chain optimization

Row 277: a precise explanation of the symbols included in the formula (for example: FR, FPC and so on…)

Row 461: „The results in Table 3.7…” – there is no Table numbered 3.7 in the article – incorrect numbering…

Discussion part of text:  It is not sufficiently explained why this study highlights the importance of strengthening the manufacturers’ role throughout the supply chain processes for the chain’s overall performance enhancement. It should be precisely demonstrated which results and how indicate such importance of producers. I propose to develop this part of the text and indicate how it affects the pursuit of optimization in the considered supply chain.

Author Response

Ms. Ref. No.: jmse-2413422

Dear Editors and Reviewers

Greetings

We thank you for the generous comments, feedback, and suggestions on the manuscript ID: jmse-2413422 for the improvement of our study.

We have specifically answered the comments from the reviewers in the tables below.

Best regards,

The authors

_____________________________

Reviewer comments

Reviewer #2

Authors response

Interesting adaptation of Bat algorithm for optimization problems in offshore fishery suplly chain.

Thank you for the feedback. Very well appreciated.

Figure 1: unclear: what elements in this diagram show that it is about integrated logistics? Or maybe it is rather an illustration of the assumptions of flows of goods and reverse flows of funds between participants

Thank you for the fruitful comment.

Figure 1 visualizes the complex relationships among participants that are involved in an integrated supply chain process. The emphasis of the relationships is placed on the material flow (including raw-fish-material and manufactured products) and cash/financial flow.

Row 113: manufactured plants or manufacturing plants? Care on accuracy of description.

Thank you for pointing out the inaccuracy.

The authors corrected the terminology to ‘manufacturing plants’.

Rows 175-176: optimal model or model used for optimization? As it is stated in the introductory part, the problem of optimization in supply chain is considered.

Thank you for fruitful comment.

The authors corrected the phrase to ‘a model of supply chain based on profit maximization’.

Rows 198-202: There is no information about keyword appropriate, according to objective of the paper such as: supply chain optimization

Thank you for the fruitful comment.

The suggested keyword of “supply chain optimization” is implied in the first two used keywords which are “supply chain model” and “optimizing model”.

Row 277: a precise explanation of the symbols included in the formula (for example: FR, FPC and so on…)

Thank you for the fruitful comment.

The authors have added an explanation of the following symbols:

FR = The total amount revenue

FC = The total amount of cost

FPC = The total amount of cost of acquiring the materials necessary for manufacturing

FMC = The total amount of cost of manufacturing

FIC = The total cost of all inventory held in stock

FTC = The total cost of transportation

FLC = The total amount of cost for labor

VC = The total amount of cost of vessel operations

Row 461: „The results in Table 3.7…” – there is no Table numbered 3.7 in the article – incorrect numbering

Thank you for the fruitful comment.

The results are referred to Table 6. The authors have corrected it in the manuscript.

Discussion part of text:  It is not sufficiently explained why this study highlights the importance of strengthening the manufacturers’ role throughout the supply chain processes for the chain’s overall performance enhancement. It should be precisely demonstrated which results and how indicate such importance of producers. I propose to develop this part of the text and indicate how it affects the pursuit of optimization in the considered supply chain.

Thank you for the fruitful comment.

The discussion in question is mainly based on the results shown in Table 6, there is a difference in aggregate profit when wholesalers (m2) and manufacturers (m3) are compared. The results imply that aggregate profit is optimized by raising the inventory of the manufacturers (m3) at $68240 in comparison to raising that of the wholesalers (m2) at $66290.

Reviewer 3 Report

This paper is a study of the supply chain of Taiwan’s offshore fishery. You evidently seek to make this supply chain more efficient and examine many links in it with a view to improving its performance. I am not qualified to comment on the mathematical modelling methods you employed, so another reviewer will have to evaluate them. My own comments are four-fold: 

(1)    You do not tell us why you are interested in this topic. Are you implying that the existing supply chain is inefficient? If so, you should explain what these inefficiencies are. In the Abstract, you say “there are challenges that hinder the offshore fishery supply chain” (lines 10-11). But you don’t explain what these challenges are or how they currently hinder the supply chain. In lines 32-34, you say “wild fishing depends on fish species, weather conditions and government regulations. These conditions have caused challenges in the offshore fishery supply chain”. In lines 498-500, you say “the fishery industry faces several challenges due to fish species availability in different seasons, weather conditions and relevant regulations, which affect the fishery supply chain”. But again, you don’t explain what these challenges are or how they affect the supply chain. Moreover, you say Taiwan’s offshore fishery is prosperous:

The Taiwanese offshore fishing industry has grown prosperously in recent years. The aquatic product market has turned export-oriented, facilitating the product supply chain to develop more maturely along with the cold chain technology development” (lines 26-28).

“Due to the abundance of the marine resources, the deep-sea fishery and offshore fishery are both prosperous in Taiwan” (lines 496-497

If it ain’t broke, why fix it?

(2)    You assert that “it is necessary to develop an optimal harvesting model under the offshore fishery supply chain evaluated based on profit maximization” (lines 13-15). But why is it necessary to construct a model based on profit maximisation? For many researchers profit maximisation is not necessarily a desirable objective, especially if it is at the expense of other values, You need to defend your assumption that profit maximisation is the one and only objective.  

(3)    You refer to interview data:

The estimation of relevant parameters was acquired through the interview data. Due to a difficulty in obtaining some confidential government information, it is hard to estimate related paramaters directly from the statistical data of the information, such as unit operating costs and revenue. To tackle such a difficulty, this study incorporates the Fisheries Agency and the Department of Environmental Biology and Fishery Sciences of National Taiwan Ocean University for assistance by interviewing the experts. The interview was focused on the potential operational performance (e.g., the probable range of operating revenues and costs) and the corresponding logistic constraints involving cold chain technology such as the capacity of facility, availability of fleet size, and frequency on vehicle scheduling. The collected data through the interviews were analyzed and processed to generate the corresponding upper and lower bounds of cost and revenue” (lines 394-404).

But you do not explain how many interviews were carried out; how the interviewees were selected; how representative were the interviewees; nor how reliable were their responses. Moreover, if you are here making use of qualitative methods in a paper which is otherwise based on quantitative methodology, you should explain and justify your mixed methods.    

(4)    Some of your ‘findings’ are banal. For instance, on lines 460-463, you say:

“The results in Table 3.7 indicate that among the involved players – raw material suppliers, wholesalers, retailers, consumers, manufacturers – that influence the overall supply chain performance, the manufacturers play a significant role in improving the model in terms of yielding the optimal profits”.

And on lines 508-510, you say:

“The findings indicate that among the key players throughout the supply chain processes, manufacturers play a crucial policy- and decision-making role in enhancing the chain performance”

                Surely it is blindingly obvious that manufacturers play a crucial role in supply chain performance?  

 Generally well-written, your paper has some infelicities of expression and some spelling errors. For example, the following passage on lines 155-159 is so dense with jargon that it is well-nigh unintelligible, and it has also misspelled two words:   

“Moreover, a hybrid model was proposed using Lagrangian Relaxation (LR) method, meta-heuristic method, bat algorithm and practicle swarm optimization method to solve the long-term production scheduling problem caused by determinisitc assumptions and level uncertainties. Such a hybrid provides a near-optimal solution”.

Author Response

Ms. Ref. No.: jmse-2413422

Dear Editors and Reviewers

Greetings

We thank you for the generous comments, feedback, and suggestions on the manuscript ID: jmse-2413422 for the improvement of our study.

We have specifically answered the comments from the reviewers in the tables below.

Best regards,

The authors

_____________________________

Reviewer comments

Reviewer #3

Authors response

This paper is a study of the supply chain of Taiwan’s offshore fishery. You evidently seek to make this supply chain more efficient and examine many links in it with a view to improving its performance. I am not qualified to comment on the mathematical modelling methods you employed, so another reviewer will have to evaluate them.

Thank you for the feedback. Very much appreciated.

(1)      You do not tell us why you are interested in this topic. Are you implying that the existing supply chain is inefficient? If so, you should explain what these inefficiencies are.

In the Abstract, you say “there are challenges that hinder the offshore fishery supply chain” (lines 10-11). But you don’t explain what these challenges are or how they currently hinder the supply chain. In lines 32-34, you say “wild fishing depends on fish species, weather conditions and government regulations. These conditions have caused challenges in the offshore fishery supply chain”. In lines 498-500, you say “the fishery industry faces several challenges due to fish species availability in different seasons, weather conditions and relevant regulations, which affect the fishery supply chain”. But again, you don’t explain what these challenges are or how they affect the supply chain. Moreover, you say Taiwan’s offshore fishery is prosperous:

The Taiwanese offshore fishing industry has grown prosperously in recent years. The aquatic product market has turned export-oriented, facilitating the product supply chain to develop more maturely along with the cold chain technology development” (lines 26-28).

“Due to the abundance of the marine resources, the deep-sea fishery and offshore fishery are both prosperous in Taiwan” (lines 496-497

If it ain’t broke, why fix it?

Thank you for the fruitful comment.

The authors have elaborated a couple of challenges mentioned in the study, as follows.

“Despite the abundance of marine resources, several challenges remain to exist to hinder the offshore fishery supply chain, including (a) dynamic environment such as demands in the local and global markets which affect fishing, production, and distribution activities in the offshore fishery industry. Fluctuations in market demands can pose challenges for supply chain management and coordination; and (b) supply and demand issues such as balancing the availability of fish species, weather conditions, and government regulations which can impact the stability and efficiency of the supply chain.”

(2)    You assert that “it is necessary to develop an optimal harvesting model under the offshore fishery supply chain evaluated based on profit maximization” (lines 13-15). But why is it necessary to construct a model based on profit maximisation? For many researchers profit maximisation is not necessarily a desirable objective, especially if it is at the expense of other values, You need to defend your assumption that profit maximisation is the one and only objective.  

Thank for the fruitful comment.

The authors have added the necessary argument to address the necessity to construct a model based on profit maximization, as follows.

“Constructing a profit-maximizing model in the offshore fishery industry is necessary for optimizing the supply chain, meeting market demand, managing resources, and addressing complex supply chain problems. Furthermore, it assesses the impacts of industrial development on cost problems and fishery management, improving coordination and efficiency. Overall, the profit-maximizing model enables efficient supply chain management in the offshore fishery industry.”

(3)    You refer to interview data:

The estimation of relevant parameters was acquired through the interview data. Due to a difficulty in obtaining some confidential government information, it is hard to estimate related paramaters directly from the statistical data of the information, such as unit operating costs and revenue. To tackle such a difficulty, this study incorporates the Fisheries Agency and the Department of Environmental Biology and Fishery Sciences of National Taiwan Ocean University for assistance by interviewing the experts. The interview was focused on the potential operational performance (e.g., the probable range of operating revenues and costs) and the corresponding logistic constraints involving cold chain technology such as the capacity of facility, availability of fleet size, and frequency on vehicle scheduling. The collected data through the interviews were analyzed and processed to generate the corresponding upper and lower bounds of cost and revenue” (lines 394-404).

But you do not explain how many interviews were carried out; how the interviewees were selected; how representative were the interviewees; nor how reliable were their responses.

Moreover, if you are here making use of qualitative methods in a paper which is otherwise based on quantitative methodology, you should explain and justify your mixed methods.

Thank you for the fruitful comment.

The authors have added an explanation to further explain the interview data, as follows.

“This study carried out one round of interview using a purposive sampling technique of professors from National Taiwan Ocean University based on their extensive years of research experience in supply chain management and logistics and the quantity of published research papers. Thus, this study uses a qualitative method by employing a complex cost function with multiple restrictions to define the relation and analysis sensitivity of quantity and price.”

(4)    Some of your ‘findings’ are banal. For instance, on lines 460-463, you say:

“The results in Table 3.7 indicate that among the involved players – raw material suppliers, wholesalers, retailers, consumers, manufacturers – that influence the overall supply chain performance, the manufacturers play a significant role in improving the model in terms of yielding the optimal profits”.

And on lines 508-510, you say:

“The findings indicate that among the key players throughout the supply chain processes, manufacturers play a crucial policy- and decision-making role in enhancing the chain performance”

Surely it is blindingly obvious that manufacturers play a crucial role in supply chain performance?

Thank for you the fruitful comment.

The authors have rewritten some parts of the discussion about the findings, as follows.

“This study utilizes the bat algorithm, known for its efficiency and adaptability to obtain near-optimal solutions. The numerical results indicate that increasing the inventory limit of the manufacturer can lead to higher profits. This suggests that suppliers can provide fishery materials to more profitable trading partners. This study focuses on the revenue-and-cost model with small-scale and basic parameters, but acknowledges the potential for more complexity in larger logistic systems.”

Generally well-written, your paper has some infelicities of expression and some spelling errors. For example, the following passage on lines 155-159 is so dense with jargon that it is well-nigh unintelligible, and it has also misspelled two words:   

“Moreover, a hybrid model was proposed using Lagrangian Relaxation (LR) method, meta-heuristic method, bat algorithm and practicle swarm optimization method to solve the long-term production scheduling problem caused by determinisitc assumptions and level uncertainties. Such a hybrid provides a near-optimal solution”.

Thank you for the fruitful comment.

The authors have cross-checked the manuscript for misspelling errors and rewritten the passage, as follows.

“Furthermore, a novel hybrid model was introduced, incorporating the Lagrangian Relaxation (LR) technique, a meta-heuristic method, the bat algorithm, and a practical swarm optimization method. This model aims to address the complex long-term production scheduling problem arising from deterministic assumptions and various levels of uncertainties. Through its integration of diverse optimization strategies, this hybrid model offers a near-optimal solution to the aforementioned problem.”

Round 2

Reviewer 1 Report

jmse-2413422-peer-review-v2
Optimizing supply chain model in Taiwan’s offshore fishery: Manufacturer’s role in raising inventory limit.

Authors have made a correction in the paper if compared to the first submission. Nevertheless, there are some suggestions to do before accepting the paper in this format.

1.             What is the innovation of the article? Is the algorithm novel? Is the problem new? Is the model original?

2.             Since the models (25, 29) are nonlinear how can we claim that the feasible space is a convex set and global optimal solution is found?

3.             Lines 9-25: Abstract should have one sentence per each: context and background, motivation, hypothesis, methods, results, conclusions. What problem did you study and why is it important? What methods did you use? What were your main results? And what conclusions can you draw from your results? Please make your abstract with more specific and quantitative results while it suits broader audiences.

4.             The manuscript is poorly structured. Why is a literature review jam-packed within the introduction and not presented as a standalone section?

5.             The structure of the introduction section is not good. It should have two separate paragraphs at its end, one of which presents the contribution and explanations of this work; and the other one outlines the coming sections.

6.             The introduction should be effective, clear and well organized. It should summarize relevant research to provide context, and explain what findings of others, if any, are being challenged or extended. Introduction must include motivation and background, literature review of recent scientific papers covering the topic and leading to the submission hypothesis based on the gap analysis of the previously published research. The most important is to state the hypothesis of your work based on the gap analysis of the previously published research. For scientific and research papers, it is not necessary to give several references that say exactly the same. Anyway, that would be strange, since then what is innovative scientific contribution of referenced papers? For each thesis state only one reference.

7.             Line 186: The text written in various Figures, particularly Figure 2, are not clear and hence not readable. All the diagrams should be re-plotted with good resolution. In addition, the Figure captions need to be modified for better understanding. Author is suggested to address this problem.

8.             The applicability of proposed method and algorithm should be discussed. It would be helpful to discuss how the proposed algorithm and method can be applied to other systems.

9.             Novelty not clear. A comprehensive table for literature survey should be presented by the authors to show the literature review based on their assumptions, methods, and results.

10.         As paper lacks to include the latest papers metaheuristic approaches, the paper must be added: - A hybrid marine predator sine cosine algorithm for parameter selection of hybrid active power filter.

11.         The current Conclusion is not a summary of the fulfilled workflow. The text is significantly similar to the Abstract which is a bad sign. In the conclusion section, please revise it and improve it by re-organizing it into one paragraph only including the suggested future work.

12.         I also recommend the authors to professionally get the paper proofread, as I have noticed sentences with typos and inappropriate choice of words.

***

 Extensive editing of English language required

Author Response

Reviewer #1

Authors response

Authors have made a correction in the paper if compared to the first submission. Nevertheless, there are some suggestions to do before accepting the paper in this format.

Thank you for the feedback. Very much appreciated.

What is the innovation of the article? Is the algorithm novel? Is the problem new? Is the model original?

Thank you for the questions

Q: What is the innovation of the article?

A: This study represents among the most recent studies in Taiwan that centers on the offshore supply chain.

Q: Is the algorithm novel?

A: Yes, this study utilizes the Bat Algorithm which is relatively new.

Q: Is the problem new?

A: Yes, the issue addressed in this study is new.

Q: Is the model original?

A: Yes, the model proposed in this study is original.

Since the models (25, 29) are nonlinear how can we claim that the feasible space is a convex set and global optimal solution is found?

Thank you for the fruitful comment

The authors have added a statement to address the claim regarding the feasible space is a convex set and global solution is found, as follows.

“While it is true that the models used in this study are nonlinear, the claim regarding the feasibility space being a convex set and the existence of a global optimal solution is found in prior studies that applied nonlinear models to find an optimal solution [1,28,29]. Although nonlinearity typically complicates the analysis, certain techniques and mathematical frameworks can be employed to verify convexity properties and establish the existence of global optima.”

Lines 9-25: Abstract should have one sentence per each: context and background, motivation, hypothesis, methods, results, conclusions. What problem did you study and why is it important? What methods did you use? What were your main results? And what conclusions can you draw from your results? Please make your abstract with more specific and quantitative results while it suits broader audiences.

Thank you for the fruitful comment

The authors have rewritten the abstract by highlighting the suggested points, as follows.

“This study delves into the offshore fishing industry in Taiwan, emphasizing the significance of the aquatic product market, supply chain development, and the maturity of cold chain technology. Taiwan’s geographical advantage of being surrounded by the sea provides a thriving environment for marine resources and migratory fish. This study is motivated by the increasing demand for diverse fish products, driven by the growing need for high-quality protein. Recognizing the importance of meeting this demand, this study aims to investigate the capacity of logistics systems and cold storage in the offshore fishery industry, particularly under conditions of uncertainty. To tackle the optimization challenges prevalent in the offshore fishery supply chain, this study employs the Bat Algorithm (BA), a meta-heuristic algorithm inspired by the remarkable echolocation behavior of bats. Additionally, a systematic literature review methodology is utilized to gather relevant articles and establish a comprehensive understanding of the study domain. This study culminates in proposing an optimized fishing model for the offshore fishery supply chain, highlighting the significance of evaluating supply chain value from a management perspective and identifying existing deficiencies and bottlenecks in current research. By focusing on optimizing the offshore fishery supply chain, this study aims to enhance the industry’s efficiency and effectiveness, providing valuable insights and recommendations to improve the capacity of logistics systems and cold storage. Furthermore, this study presents the results of the BA, showcasing its effectiveness in approaching the optimization challenges, thereby validating its utility for the offshore fishery industry. Sensitivity analysis reveals the potential for higher profits by raising the inventory limit of the manufacturer, enabling the supplier to provide materials to more profitable trading partners. While this study is based on a revenue-and-cost model, it acknowledges that the objective and constraints would become more complex in varying logistic system circumstances. Future study aims to expand the scale of the model and incorporate practical cases to further enhance its applicability.”

The manuscript is poorly structured. Why is a literature review jam-packed within the introduction and not presented as a standalone section?

Thank you for the fruitful comment

The authors have dedicated the Literature Review to a separate section from the Introduction.

The structure of the introduction section is not good. It should have two separate paragraphs at its end, one of which presents the contribution and explanations of this work; and the other one outlines the coming sections.

Thank you for the suggestion

The authors have ensured that there are two separate paragraphs presenting the study’s contributions and the outlines of the coming sections at the end of the Introduction section.

The introduction should be effective, clear and well organized. It should summarize relevant research to provide context, and explain what findings of others, if any, are being challenged or extended. Introduction must include motivation and background, literature review of recent scientific papers covering the topic and leading to the submission hypothesis based on the gap analysis of the previously published research. The most important is to state the hypothesis of your work based on the gap analysis of the previously published research. For scientific and research papers, it is not necessary to give several references that say exactly the same. Anyway, that would be strange, since then what is innovative scientific contribution of referenced papers? For each thesis state only one reference.

Thank you for the fruitful comment

The authors have pointed out the points in question.

For example:

Regarding motivation and background:

“The increasing demand for various fish products is driven by the growing need for high-quality protein. Moreover, wild fishing activities are dependent on factors such as fish species, weather conditions, and government regulations [2]. Despite the abundance of marine resources, several challenges persist, impeding the efficiency of the offshore fishery supply chain. These challenges can be categorized into two main areas: (a) the dynamic nature of the environment, including fluctuating demands in local and global markets, which directly impact fishing, production, and distribution activities within the offshore fishery industry; and (b) supply and demand issues, encompassing the delicate balance between fish species availability, weather conditions, and compliance with government regulations. These factors collectively affect the stability and efficiency of the supply chain.”

Regarding previous studies’ different findings:

“Prior studies have attempted to identify the various influential factors in the supply chain where purchased quantity, fish weight, sale price, inventory, and transportation are identified as the most influential factors in the whole supply chain model [3-5]. However, these studies focused on different assumptions such as how three types of supply chain are able to coexist and interact with each other, how information disclosure might infer the demand received by the downstream players, and how carbon footprint is considered and estimated in the capture seafood industry management. This study, nevertheless, focuses on proposing a profit-maximizing model to reach an efficient supply chain management in the offshore fishery industry.”

Line 186: The text written in various Figures, particularly Figure 2, are not clear and hence not readable. All the diagrams should be re-plotted with good resolution. In addition, the Figure captions need to be modified for better understanding. Author is suggested to address this problem.

Thank you for the fruitful comment.

The authors have ensured the readability of the figures, especially Figure 2.

The applicability of proposed method and algorithm should be discussed. It would be helpful to discuss how the proposed algorithm and method can be applied to other systems.

Thank you for the fruitful comment.

This study explores the use of metaheuristic algorithms, including particle swarm optimization, harmony search, and firefly algorithm, as powerful methods for solving challenging optimization problems. Specifically, it focuses on the Bat Algorithm, a heuristic algorithm. The Bat Algorithm draws inspiration from the sophisticated echolocation abilities of bats, which use sound pulses and echoes to navigate and locate prey in the dark. By observing the behavior and characteristics of microbats, this study formulates the Bat Algorithm based on three key aspects of the echolocation process. These aspects include the use of echolocation to sense distance and encode positions, the flying behavior of bats with varying frequency, wavelength, and loudness to search for prey, and the adjustment of pulse emission based on proximity to the prey source. This study assumes a minimum loudness value and utilizes the relationship between sonic velocity, wavelength, and frequency to inform the algorithm's operation. Nevertheless, the different contexts where the Bat Algorithm was applied include a control of energy consumption and tracking accuracy in robots [30], an exploration of buried ore and mineral targets parameters [31], and an assessment of multi-classification problem of Cardiotocography [32]. Meanwhile, the application of the algorithm in the context of offshore fishery supply chain is scarce in the literature.

Novelty not clear. A comprehensive table for literature survey should be presented by the authors to show the literature review based on their assumptions, methods, and results.

Thank you for the suggestion.

The authors have added the suggested table. Please refer to Table 1.

As paper lacks to include the latest papers metaheuristic approaches, the paper must be added: - A hybrid marine predator sine cosine algorithm for parameter selection of hybrid active power filter.

Thank you for the feedback.

The authors will consider including the suggested information regarding other metaheuristic approaches in the future study.

The current Conclusion is not a summary of the fulfilled workflow. The text is significantly similar to the Abstract which is a bad sign. In the conclusion section, please revise it and improve it by re-organizing it into one paragraph only including the suggested future work.

Thank you for the fruitful comment

The authors have rewritten the conclusion and composed it into a single paragraph.

I also recommend the authors to professionally get the paper proofread, as I have noticed sentences with typos and inappropriate choice of words.

Thank you for the recommendation

The manuscript has been checked for grammatical errors and other mistakes.

Round 3

Reviewer 1 Report

jmse-2413422-peer-review-v3

Optimizing supply chain model in Taiwan’s offshore fishery: Manufacturer’s role in raising inventory limit.

The revised paper has not been improved, and I think the authors' attitude to the manuscript is rather casual. I do not see anything new such as the appropriateness, novelty and general significance. There are also several technical content and quality issues. I regret to recommend to reject this paper.

***

·    A proof reading by a native English speaker or proofreading service should be conducted to improve both language and organization quality. The same should be supported by a proof-read certificate.